# Impact-induced changes in source depth and volume of magmatism on Mercury and their observational signatures

Sebastiano Padovan [1], Nicola Tosi [1,2], Ana-Catalina Plesa [1] & Thomas Ruedas [1,3]

Mercury's crust is mostly the result of partial melting in the mantle associated with solid-state convection. Large impacts induce additional melting by generating subsurface thermal anomalies. By numerically investigating the geodynamical effects of impacts, here we show that impact-generated thermal anomalies interact with the underlying convection modifying the source depth of melt and inducing volcanism that can significantly postdate the impact depending on the impact time and location with respect to the underlying convection pattern. We can reproduce the volume and time of emplacement of the melt sheets in the interior of Caloris and Rembrandt if at about 3.7–3.8 Ga convection in the mantle of Mercury was weak, an inference corroborated by the dating of the youngest large volcanic provinces. The source depth of the melt sheets is located in the stagnant lid, a volume of the mantle that never participated in convection and may contain pristine mantle material.

[1] Department of Planetary Physics, German Aerospace Center, 12489 Berlin, Germany. [2] Department of Astronomy and Astrophysics, Technische Universität Berlin, 10623 Berlin, Germany. [3] Institute of Planetology, University of Münster, 48149 Münster, Germany. Correspondence and requests for materials should be addressed to S.P. (email: sebastiano.padovan@dlr.de)

The crust of Mercury is mostly volcanic in origin and relatively young, having its oldest units being dated to 4.1 Ga[1,2]. The youngest surface units are represented by smooth plains, which cover about 27% of the surface[3]. The four largest smooth plain units (Northern plains, plains within and around the Caloris basin, and the Rembrandt basin interior plains) have been emplaced at around 3.7 Ga[3–5]. Recent dating of additional smooth plain units places the termination of widespread volcanic events on Mercury at about 3.5 Ga[6]. It has been argued that subsequent extrusive volcanism might have been hindered by the accumulation of compressive stresses in the lithosphere induced by the contraction of the planet[6]. However, the contraction is a result of cooling and thus of decreasing temperature and melt production in the interior (see below). Moreover, the numerous thrust faults seen on the surface are the surficial manifestation of the contraction of the planet and correspond to weak zones in the lithosphere. They would have provided conduits for magma ascent, if enough magma had been produced in the mantle[7]. Therefore, the end of widespread volcanism on Mercury at 3.5 Ga indicates that by that time the amount of melt associated with convection in the mantle of Mercury was diminishing or negligible. Post-MESSENGER thermal and thermo-chemical evolution models[8,9] are roughly consistent both with the duration of the volcanic activity observed on Mercury[6] and with the inferred volume of its crust[10]. However, the linkage between the internal dynamical processes creating the crust and the geochemically varied surface composition has not been fully clarified. The following processes have been proposed to interpret the observed geochemical terranes: a heterogeneous mantle generating magmas of different composition in different areas[11–13]; a rapidly cooling mantle where temperature and pressure in the melt source regions change rapidly in time, thus producing melts of different compositions[13,14]; and a homogeneous mantle whose composition evolves in time due to the cumulative depletion of the magma source[12,15].

In addition to convection in the mantle, large impact events, as recorded by the large basins observed on the surface of Mercury[16], may induce melting by three different processes. First, the release of the shock-pressure associated with the impact can increase the temperature above the solidus in a significant volume in the proximity of the impact location and create shock-melt that forms a melt pool, which rapidly solidifies possibly undergoing igneous differentiation[17]. Second, the pressure field under a newly formed basin changes due to material being excavated and ejected outside of the crater rim. The local modification of the lithostatic pressure field under the crater depresses the solidus in the mantle and may induce in situ decompression melting[18]. Third, the interaction of the impact-induced thermal anomaly in the subsurface with the preexisting temperature field induces convective motions potentially followed by additional postimpact melting[18,19]. Both shock-melting and in situ decompression melting are processes that happen instantaneously from a geological perspective. Post-impact melting operates on

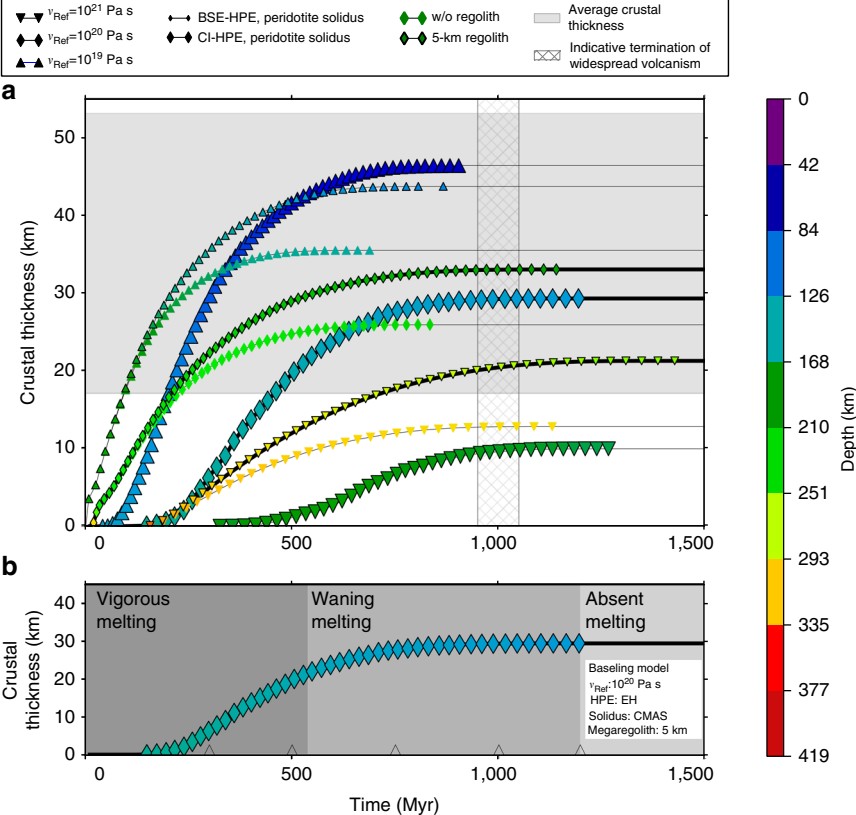

**Fig. 1** Crustal thickness from convective melting as a function of time. **a** Each set of data corresponds to different values of the reference viscosity (symbol shape), initial amount of heat-producing elements in the mantle (HPE) and solidus parameterization (symbol size), and presence of a surficial megaregolith layer (indicated by a black contour). Symbol color indicates the average depth of the melt source region according to the color bar. For each model, the rightmost symbol plotted corresponds to the termination of convective-melt production. Models consistent both with the inferred crustal thickness[10] (gray horizontal band) and the duration of volcanic activity[6] (hashed vertical band) are drawn with a thicker black line. **b** The three mantle melting regimes are shown for the baseline model, whose parameters are indicated in the white box. The triangles along the time axis indicate the times of impact for the simulations shown in Figs. 4 and 5

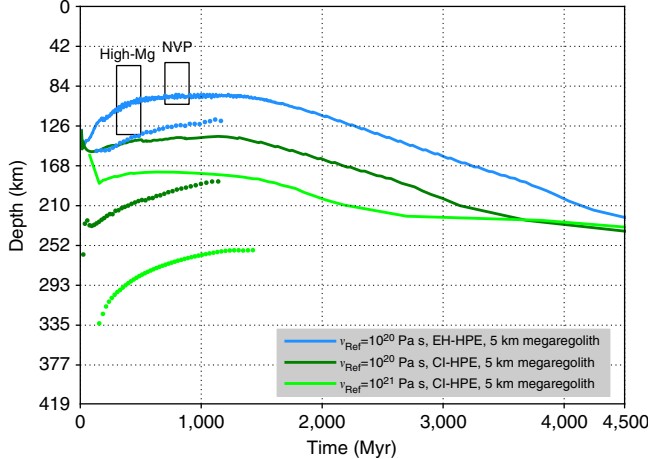

**Fig. 2** Stagnant lid evolution. Temporal evolution of the location of the bottom of the stagnant lid (solid lines) and of the source depth of convective melt (dots) for the models compatible with the thickness of the crust and the duration of volcanic activity (thick black lines in Fig. 1a). The black boxes indicate the inferred thickness of the lid at the time of emplacement of the High-Mg region and the northern volcanic plains (NVP)[13]. The model in blue satisfies the lid thickness constraint and is selected as the baseline model (Fig. 1b)

convective timescales—from few to hundreds of millions of years—and thus postdates the impact event[18].

All major basins on the surface of Mercury show signs of volcanic infillings that postdate the basin-formation event. The temporal delay and the volume of the volcanic infilling have been quantified for a small number of craters, including the young large basins Caloris and Rembrandt[16,20–23]. Such basins are also associated with crustal thinning[24,25]. Here we model the thermal evolution of Mercury and include the effects of large impacts on magmatic activity in the interior. We can qualitatively explain the crustal thinning associated with young large basins and quantitatively reproduce the volume and time of emplacement of the volcanic melt sheets of Caloris and Rembrandt. These results depend on the thermal state of the mantle at the time of the impacts and are compatible with the volcanic history of Mercury as inferred from the dating of the surface. Interestingly, the source depth of the material in the melt sheets of Caloris and Rembrandt is located in the stagnant lid, a volume of the mantle that never underwent partial melting and thus contains potentially pristine mantle material. Such material, otherwise not sampled by convection-induced partial melting, may explain the compositional signature associated with the interior of Caloris[14,15,26].

## Results

**Thermal evolution models.** With the convection code GAIA[27], we first compute thermal histories of Mercury by adopting commonly accepted values for convection parameters and by varying the value of the reference viscosity, of the thickness of the surficial megaregolith layer[28,29], and of the abundance of heat-producing elements (HPE) in the mantle. For the HPE, we employ two models in order to encompass their unknown bulk abundance in Mercury. These models are based on the abundances of potassium, thorium, and uranium in enstatite chondrites[30] (EH-HPE) and carbonaceous chondrites[31] (CI-HPE). We also tested the bulk silicate Earth model of Lyubetskaya and Korenaga[32], but it was found to be incompatible with the available constraints (see below). While the adopted models do not accurately represent the bulk silicate Mercury, they cover a

conservative range that likely encompasses the actual HPE abundances. HPE abundances and associated heat productions are listed in Supplementary Table 1. HPE are uniformly distributed in the mantle and no primordial crust is assumed. The entire set of parameters used in the simulations is listed in Supplementary Table 2.

During the evolution we track melt (i.e., crustal) production and calculate the source depth of the melt produced as a function of time. We account for the mantle depletion of HPE when extracting melt (see Methods section). Partial melts in Mercury's mantle are buoyant over the entire mantle depth range, and they always contribute to the building of the crust[33]. To compute melt production we consider two different melting curves: an anhydrous peridotite solidus[34] for the CI-HPE models and an iron-free CMAS solidus corrected for sodium oxide, as appropriate for Mercury[13], for the EH-HPE models (the acronym CMAS comes from the simplified system of oxides—CaO, MgO, $Al_2O_3$, $SiO_2$—used to represent major-element mantle chemistry). These melting curves and their parameterizations are reported in Supplementary Fig. 1 and Supplementary Table 3, respectively. Different assumptions lead to variations in the total crust produced, in the timing of crust production, and in the characteristic source depth of the melt associated with convection (Fig. 1a). Other things being equal, a larger amount of HPE in the mantle is associated with a thicker crust since the increased heat production leads to higher temperatures and thus a larger amount of melt (Supplementary Fig. 2a). A megaregolith layer acts as a thermal blanket that prevents efficient heat extraction from the mantle, thus favoring higher temperatures and increased melt and crustal production. Crustal production is concentrated in the early phases of evolution when the mantle is vigorously convecting. The Rayleigh number, a measure of the strength of convection, is inversely proportional to the reference viscosity. Thus a higher reference viscosity corresponds to a less vigorously convecting mantle and to a thinner crust.

Independent of the choice of parameters, crustal production is very rapid in the early stages of the evolution and is completed between about 500 Myr and 1.5 Gyr. The source depth of the melt is deeper for higher reference viscosities (consistent with classical boundary layer theory, see Methods section) and shallower in the presence of a megaregolith layer. It is roughly independent of the amount of HPE in the mantle (Supplementary Fig. 2a). The source depth of the melt associated with solid-state convection is roughly constant during the entire evolution of the planet, as the limited variation of the color of the symbols for each curve in Fig. 1a illustrates (see also Supplementary Figs. 3 and 4).

**Baseline model.** A subset of the models plotted in Fig. 1a are compatible both with the inferred thickness of the crust[10,25] (those having a crustal thickness within the horizontal gray band) and with the duration of widespread volcanism on the planet[6] (those whose volcanic activity extends to the right of the vertical hashed band). These are plotted with a thick black line. For this subset of models, we compute the evolution of the stagnant lid thickness (Fig. 2). The two black boxes in Fig. 2 indicate the location of the lid at the time of the formation of the high-magnesium region (High-Mg), which comprises one of the oldest terrain observed on Mercury[1], and of the Northern plains[13] (NVP), as inferred from numerical modeling and laboratory experiments[13]. For the vertical extension of the boxes, we use data from Namur et al.[13] (their Fig. 7), and for the horizontal extent we use data from Marchi et al.[1] and Ostrach et al.[35]. In Fig. 2, the model corresponding to a reference viscosity $\nu_{Ref} = 10^{20}$ Pa s, an abundance of HPE based on the enstatite chondrites[30], and a

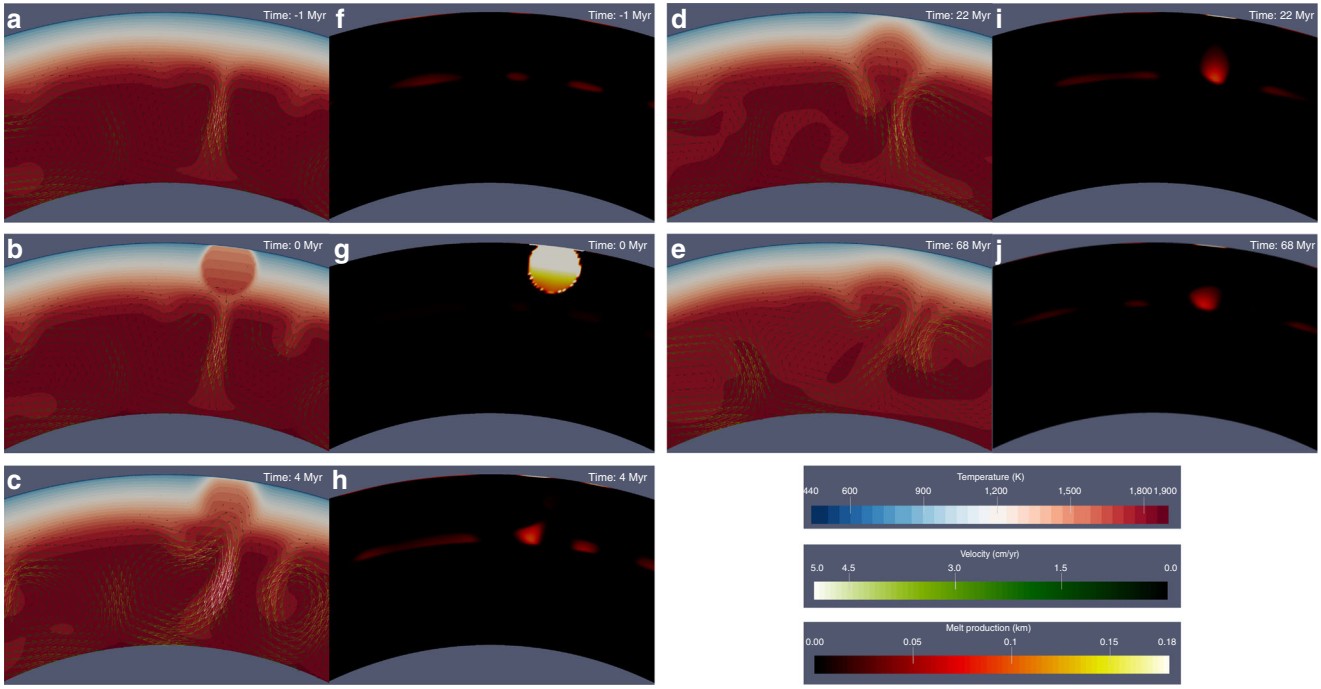

**Fig. 3** Impact-induced convective currents and melt production. The images pertain to the baseline model (Fig. 1b) in the event of an impact happening at $t$ = 300 Myr and producing a Caloris-sized basin. **a–e**: Temporal evolution of the temperature field (background) and of the magnitude of the velocity field (arrows). **f–j**: Temporal evolution of the melt production, which is expressed in equivalent crustal thickness. The outermost layer refers to the column-integrated value. For the simulation shown here, the impactor hits the target with a velocity of 42.5 km/s and an impact angle of 45°. The impact event corresponds to 0 Myr. The thermal anomaly warps the isotherms and convective currents, which generate melt, develop in the region close to the impact but with a certain delay with respect to the impact event. Enhanced melt production persists below the impact location for >100 Myr. The thermal anomaly is completely absorbed after about 180 Myr (not shown)

5-km-thick megaregolith layer fits the inferred location of the lid. Therefore, we select it as the baseline model. For the baseline model, we compute the total thermal contraction of the planet (see Methods section) following the mantle peak temperature (Supplementary Fig. 4). The total contraction is 9.9 km, of which 4.7 km are accommodated by mantle cooling. The total value is slightly in excess of the 7 km inferred from the analysis of surface contractional features[36]. However, due to unfavorable illumination geometry 7 km is likely an underestimate of the total contraction[36]. In addition, part of the radius decrease is accomodated without any manifestation in the geologic record[37]. This "invisible" component can be as large as 2.5 km for Mercury, potentially bringing the total inferred contraction to 9.5 km[37]. The set of parameters characterizing the baseline model can be regarded as representative of Mercury. However, the details of the model do not affect our analysis of the effects of large impacts. In the Supplementary Section, we provide results obtained using a different baseline model (Supplementary Figs. 2b and 5).

**Melting regimes.** The large expanses of smooth plains are the last events of massive widespread volcanism on Mercury. Their emplacement terminates at about 3.5 Ga[6]. If the smooth plains are volcanic eruptions with a 10% extrusive-to-intrusive ratio, their total volume represents about 16% of the total volume of the crust (see Methods section). We use this estimate to define three regimes of melt production in the mantle. Initially the mantle is convecting vigorously, producing melt which rapidly contributes to the building of the silicate crust (vigorous melting regime). The time when the volume of the crust equals the final crustal volume minus the volume of the melt associated with the smooth plains, as estimated above, defines the beginning of the second, waning melting regime. The epoch following the last melting event (i.e.,

once the thickness of the crust reaches its final value and melt production ceases in the mantle) defines the beginning of the final phase or the absent melting regime. The building of the crust as the cumulative result of partial melting in the mantle is a continuous process and the definition of the three regimes is somewhat arbitrary since it is based on an assumed extrusive-to-intrusive ratio and thickness of the extruded part for the smooth plains. However, it provides a simple classification criterion to interpret the geodynamical effects of large impacts discussed below. The three regimes for the baseline model are illustrated in Fig. 1b.

**Effects of large impacts on convection.** We account for the effects of large impact events on the dynamics of the mantle by using scaling laws to evaluate the thermal anomaly resulting from the release of the impact shock-pressure[19,38,39]. The thermal perturbation upon impact brings the temperature in a volume around the impact location well above the solidus, consistent with the formation of a melt pool[17]. The solidification of the melt pool is rapid compared to convection timescales[17,40], and therefore we truncate the temperature at the local solidus[19,39,41]. We employ the same approach as detailed in Roberts and Barnouin[19]. The population of impactors on Mercury has a broad distribution of encounter velocities, between about 15 and 60 km/s, with a mean encounter velocity of 42.5 km/s[42]. The most probable impact angle is 45° (ref.[42]). Since the scaling laws are referred to the vertical component, the velocity is corrected accordingly[19] and a Caloris-sized basin requires an impactor with a mean encounter velocity and a diameter of about 92 km (see Supplementary Fig. 6). The thermal anomaly induced by an impact warps the isotherms and produces convective currents that enhance melt production and change the

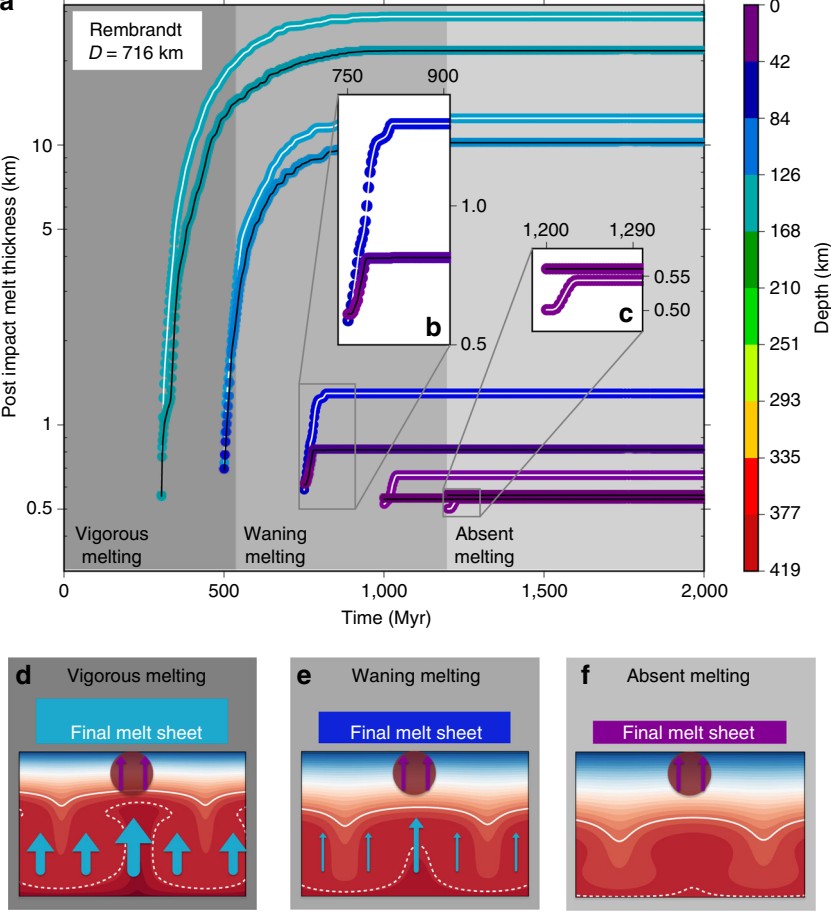

**Fig. 4** Melt production following an impact forming a Rembrandt-sized basin. **a** Results using the baseline model (Fig. 1b) for the impact occurring at 300, 500, 750, 1000, and 1197 Myr. The impactor has a diameter of 37 km and hits the target with a velocity of 42.5 km/s at an impact angle of 45°. For each epoch, the cases for the impact occurring over an upwelling and a downwelling are plotted. The impact over the downwelling (black line) initially extracts shallower material with respect to the impact over the upwelling (white line). Differences in the total amount of melt for impacts over a downwelling vs. impacts over an upwelling depend on the size of the basin with respect to the size of the convection cells in the mantle (Fig. 6). The gray background indicates the melting regime as in Fig. 1b, which are illustrated in **d**–**f** at the bottom of this figure. **b**, **c** Zoom on the first few tens of Myr after the impact events at 750 and 1197 Myr. For impacts happening in the vigorous melting regime and early in the waning melting regime, the depth of the source region for the postimpact melt is rapidly controlled by the contribution of the convective melting (**d**, **e** below). For the impact at 1197 Myr the contribution of convection melting is almost absent, and the melt is the result of partial melting in the shallow mantle (**f**, below). The postimpact melt thickness (y-axis) represents an upper bound for the melt sheet thickness within the basin. **d**–**f** For the three melting regimes, the cartoons illustrate the temperature field in the mantle (background red/blue field), the impact-induced thermal anomaly (spherical red shape), and the melting contribution both from convection (azure arrows) and from postimpact melting (violet arrows). The white lines represent a cold (solid) and a hot (dashed) isotherm. The area of the arrows qualitatively indicates the amount of melt produced. More melt production is associated with upwellings (see also Supplementary Fig. 3) The thickness and source depth of the final melt sheet depends on the relative contribution of the two sources of melt and is represented by the boxes labeled "Final melt sheet"

vertical extent of the partial melt zone in the volume of the mantle below the impact. We illustrate this effect in Fig. 3, which refers to the baseline model in the case of a Caloris-forming impact occurring at 300 Myr. The dynamical effects of the impact last for about 100 Myr.

For the baseline model, we simulate an impact at five different times, representative of the three regimes of melting defined above and shown in Fig. 1b. These times are 300 and 500 Myr (vigorous melting), 750 and 1,000 Myr (waning melting), and 1197 Myr (absent melting). For each of these times, we simulate two events corresponding to impacts occurring over a downwelling and an upwelling. We perform two sets of simulations using impact parameters compatible with the creation of a Rembrandt-sized basin (impactor diameter 37 km, impactor velocity 42.5 km/s, basin diameter 716 km) and a Caloris-sized basin (impactor diameter 92 km, impactor velocity 42.5 km/s, basin diameter 1550 km). For these two basins, estimates of the

time of emplacement and volume of the interior volcanic plains are available[16,20,22,43]. Starting at the time of the impact event, we keep track of the amount and source depth of the melt produced in the volume of the mantle below the final (i.e., observed) impact basin. This melt, if erupted, contributes to the formation of the melt sheet (Figs. 4 and 5).

**Postimpact eruptions.** Following a large impact event, two processes favor the subsequent eruption of magma at the surface at very high ratios of extrusive-to-intrusive volcanism. First, any preexisting crust is largely removed and only partly melted and mixed with upper mantle material in the melt pool[44,45]. Thus the possible neutral buoyancy horizon created by a low-density crust[46], which would cause rising melt to stall at depth, is completely or largely cancelled. Second, the fractured lithosphere in the region below and around the impact site[47] allows for an easier

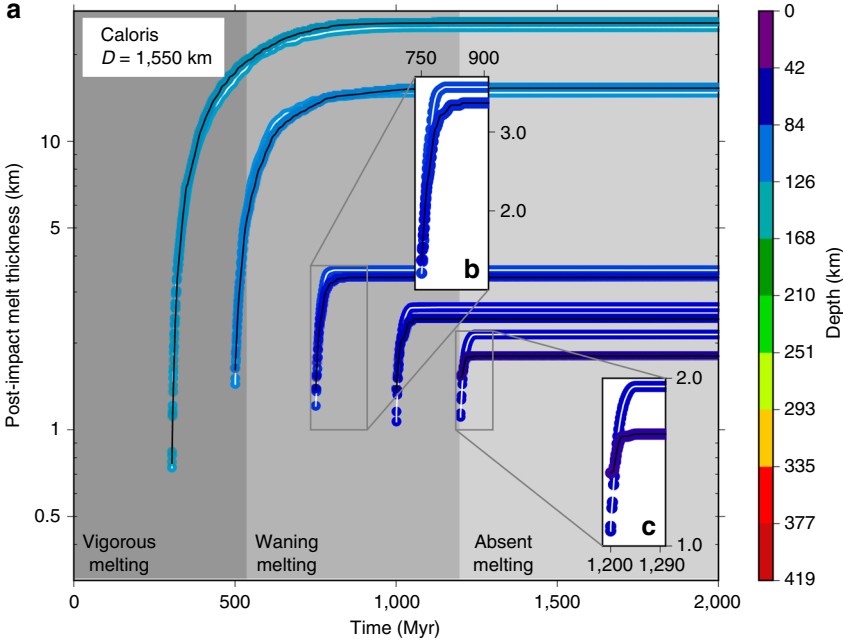

**Fig. 5** Melt production following an impact forming a Caloris-sized basin. **a** As in Fig. 4a but for an impactor with a diameter of 92 km, compatible with the formation of a Caloris-sized basin. **b, c** Zoom on the first few tens of Myr after the impact events at 750 and 1197 Myr. The larger impactor induces melting at slightly higher depths (colors) with respect to the case of Rembrandt (Fig. 4b, c)

upward migration of the melt. It is natural to expect volcanic eruptions to occur within the basin and to postdate the impact event, as long as the two conditions described above are met. All major basins on Mercury show evidence of volcanism postdating the basin formation event[16]. Therefore, conduits for magma ascent exist for geologically long time intervals[4] (tens to hundreds of millions of years). The relative size of the basin with respect to the underlying convection pattern influences the difference in the final amount of postimpact melt for impacts located over an upwelling and over a downwelling. If the size of the basin, compared to the underlying convection pattern, encompasses more than one convection cell, the final amount of melt produced is similar. Caloris is representative of this scenario (Fig. 5). For basins smaller than the typical convection cell, the influence of impact location is more pronounced, as in the case of Rembrandt (Fig. 4a). This observation can explain the differences in the postimpact melt thickness associated with impacts over upwellings and downwellings in Figs. 4 and 5. This conclusion is not affected by the cylindrical rescaling adopted[48] in our simulations, since for the very thin mantle of Mercury the aspect ratio of the convective cells is roughly the same for 2D spherical, 2D cylindrical, and 3D simulations[8,19]. The effect of impact location and basin size on the final amount of melt is illustrated in Fig. 6. The temporal delay between the basin formation and the last volcanic material extrusion also depends upon the interplay between the impact anomaly and the underlying mantle dynamics, as the impact event in the absent melting regime illustrates (Fig. 4c). An impact occurring over a downwelling, a region where cold and negatively buoyant material descends into the mantle, results in an almost immediate melting event that extracts material from very shallow depths. When the impact is localized over an upwelling, the thermal anomaly revives the upwelling, which was not producing melt at the time of the impact, and generates additional melt for an interval of time of about 70 Myr.

**Observational signatures of impacts as a function of time.** The impact-induced thermal anomaly generates melt in the shallow

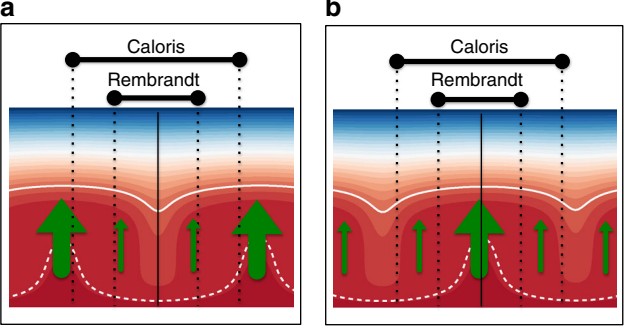

**Fig. 6** Convection pattern and basin size. In each panel, the vertical solid black line indicates the impact location, which correspond to a downwelling (**a**) and an upwelling (**b**). The background red/blue field represents the temperature field in the mantle. The white lines represent a cold (solid) and a hot (dashed) isotherm. Green arrows represent convection melting, and their area qualitatively indicates the amount of melt produced, which is larger for upwellings, where hot positively buoyant material rises, and almost absent for downwellings, where cold negatively buoyant material sinks (see also Supplementary Fig. 3). For a basin with a diameter comparable to or larger than the size of a convective cell (e.g., Caloris), the location of the impact has a minor influence on the total amount of postimpact melt produced, since the basin encompasses more than a single convective cell (vertical black dashed lines). For a smaller basins (e.g., Rembrandt), more melt is associated with the impact happening over an upwelling. Compare upwelling and downwelling cases in Figs. 4 and 5

mantle and subsequently modifies the melt production in the mantle (Fig. 3). For early impacts, when the mantle is in the vigorous melting regime and the melt production rate (i.e., the slope of the curve in Fig. 1b) is high, this shallow melt is rapidly masked by large volumes of convection-related melt that is generated at depth in the mantle, mostly in association with hot plumes (Figs. 4 and 5 and Supplementary Fig. 3). In terms of source depth, the melt erupted in the basin rapidly loses any

signature of the shallow, impact-induced melt, to resemble melt erupted at the same epoch in regions unaffected by the impact. Indeed, the source depth for the postimpact melt in the vigorous melting cases in Figs. 4 and 5 is basically undistinguishable from the convective source depth (Supplementary Fig. 3). Based on the large amount of melt produced in the early phases of the evolution, old basins should correspond to only minor, if any, crustal thinning. Crustal thickness maps corresponding to the old large basins Matisse-Repin (location −24.4 °N, 285.1 °E, diameter 887 km) and Calder-Hodgkins (location 17.1 °N, 21.7 °E, diameter 1460 km) are consistent with this expectation[24,25]. As the mantle cools, the melt production associated with convection decreases (waning melting regime) and gradually, in the aftermath of an impact, the shallow impact-related melt dominates the volume budget of the melt sheet (Figs. 4 and 5). The total amount of melt produced decreases by about an order of magnitude and the crustal thinning associated with the formation of the basins is likely preserved. Indeed, the interior of Caloris is associated with the thinnest crust on Mercury[24,25]. Once convection stops producing melt (absent melting regime), the melt forming the melt sheet would only come from the shallow mantle (Figs. 4c, f and 5c).

**A model for the melt sheets of Caloris and Rembrandt**. The emplacement of the Caloris interior plains postdates the Caloris impact event, dated at about 3.7–3.8 Ga[4,5,20], by 100–200 Myr[4,20]. This timing is compatible with the prediction of our model, according to which, depending on the impact location, melt is produced for about 60–100 Myr following the impact (Fig. 5b). From the basin stratigraphy, the volume of the Caloris interior plains has been inferred to be in the range $(3.2–5.2) \times 10^6 \, km^3$ (ref. [43]), corresponding to an average thickness of the melt sheet between about 1.7 and 2.8 km. The total amount of melt predicted by our model would correspond to a melt thickness of about 3.6 km (inset Fig. 5b). Thus we can match the observed thickness for extrusive-to-intrusive ratios in the range 50–82%. The Rembrandt basin (diameter 716 km) has a history similar to Caloris, forming at about 3.8 Ga and having its interior covered by volcanic plains that postdate the basin formation by 100–200 Myr[22]. Using the baseline model and simulating an impact appropriate for a Rembrandt-sized basin (impactor with an encounter velocity of 42.5 km/s and a diameter of 37 km), we estimate that the amount of postimpact melt produced has a thickness in the range 0.8–1.3 km, and it is emplaced in an interval of time between 30 and 70 Myr, depending on the impact geometry (Fig. 4b). Similarly to the case of Caloris, a high extrusive-to-intrusive ratio, in the range 28–64%, can explain the observed thickness of 360–520 m inferred by analysis of the crater statistics of the Rembrandt plains[22].

Our model predicts prolonged volcanic activity, from few tens to about a hundred million years, following the Caloris and Rembrandt impact events. This result is consistent with the inferred delay between the formation of the basins and the emplacement of the interior melt sheets. However, the exact timing is difficult to measure with high accuracy, since the error associated with estimates of this kind of delays is close to the value of the delay itself[49]. The large extrusive-to-intrusive ratio required to match the observed volume of the melt melt sheets within the two basins is expected, given the postimpact crustal removal and fracture of the lithosphere. These results are based on assuming the average velocity (42.5 km/s) for the impactors forming the basins. Slower impactors produce larger thermal anomalies[19]. We tested the effect of a Caloris-forming impact using an impactor with 15 km/s encounter velocity, consistent with the lower end of impactor velocities at Mercury[42]

(the corresponding impactor diameter is 164 km, Supplementary Fig. 6). For this end-member scenario, volcanic activity within the basin lasts for 75–110 Myr and an extrusive-to-intrusive ratio in the range 36–66% can explain the observed melt sheet thickness.

## Discussion

The formation of a stagnant lid is characteristic of a single-plate planet like Mercury. Its evolution depends on the assumed parameters (Fig. 2), but a number of general considerations are valid throughout the parameter space. First, independent of the initial conditions of the model, the growth of the lid thickness is initially very rapid (Fig. 2 and, e.g., ref. [8]). Second, at the onset of convection, the magnitude of the flow velocity of the mantle is larger at depth and close to zero in the upper mantle. We compared the evolution of the lid thickness for the baseline model and for the same model but with the initial temperature profile set at the solidus (Supplementary Fig. 7). We performed a similar calculation using a different baseline model (Supplementary Fig. 2b). The hotter initial conditions correspond to a thinner lid thickness (Supplementary Fig. 8). However, the values of velocities in the upper mantle quickly reach values on the order of $10^{-2}$ cm/year (corresponding to distances of about 10 km per hundred Myr), while the lid thickness grows several tens of kilometers thick. Thus the majority of mantle material in the lid never participates in the convection and in related partial-melting events.

The source depth of the volcanic infilling of large impact basins becomes shallower with time (Figs. 4 and 5). For the baseline model, a comparison of the source depth of the melt sheet for a Caloris basin forming at 3.75 Ga (Fig. 5b) with the minimum depth of the lid (>90 km, solid blue line in Fig. 2) shows that the source depth of the melt sheet is shallower than or comparable to the thickness of the lid. Therefore, the Caloris interior plains contain substantial contributions from partial melting of volumes of the mantle not sampled by convection-induced partial melting and composed of pristine mantle material, the products of the solidification of the initial postimpact melt pool, or both. A similar conclusion can be drawn for the plains within the Rembrandt basin (Fig. 4b). In general, the thicker the stagnant lid, the stronger is the contrast between the source depth of the melt sheet and the source depth of convective melt (in Supplementary Fig. 5, we present results similar to Fig. 5 obtained using a different baseline model).

Mineralogical models of the surface of Mercury show that the Caloris interior plains are peculiar in being the most plagioclase-rich area found on the surface[14]. Such mineralogical composition has been interpreted as being the result of tertiary crustal formation or of plagioclase floatation in a large melt pool[14]. Hydrocode simulations of large impacts on Mercury show that the crust present before the impact is almost entirely removed[45,50]. This scenario cannot be easily reconciled with a tertiary crust origin for the plagioclase. On geological timescales, the melt pool freezes rapidly[17,40] and a floatation origin for the plagioclase would not be consistent with the inferred delay of the emplacement of the melt sheet. In addition, due to the low iron content of the mantle of Mercury the buoyancy of plagioclase in liquid melts is reduced and its floatation unlikely[33,51]. The observed melt sheet in the interior of Caloris covers a layer of dark material[43]. Graphite is the only buoyant mineral in Mercurian melts and the low-reflectance material might represent a floatation crust formed during the solidification of the melt pool[33]. The postimpact melting model can explain the delay and volume of the melt sheet covering the dark material. However,

predicting the mineral composition of postimpact melts using a convection code would require a level of complexity—two-phase flow dynamics combined with stability of liquid and solid phases at extremely high spatial resolutions—which, to our knowledge, no global convection code is capable of doing at the present time. In addition to its forsterite-rich mineralogy, the composition of the plains in the interior of Caloris differs from that of other large volcanic units with approximately the same age but not associated with large impact basins[5,14,15,26]. Our model indicates that from a geodynamical point of view two possible components of the Caloris interior plains are partial melts of pristine mantle material and partial melts of the solidified melt pool. These components would only be relevant for young large impact basins, of which Caloris is the only example in the northern hemisphere. Their potential role in generating the unique mineralogy and composition of the Caloris plains should thus be investigated. We make similar predictions for the interior plains of Rembrandt, for which, however, detailed geochemical analyses are missing due to the low-resolution data in the southern hemisphere. Better data coverage of the southern hemisphere to be provided by the forthcoming BepiColombo mission[52] will help to further validate our model.

A large high-Mg region with an approximate diameter of 3000 km, apparent in geochemical maps of Mercury, was tentatively identified as the site of an ancient impact basin[15]. Our model predicts that the source region of postimpact melt within ancient impact basins would be indistinguishable from convective melt originating in regions of the mantle unaffected by large impacts (Figs. 4 and 5). Enhanced melt production in the mantle below the impact location (Fig. 3) is compatible with the generation of Mg-rich magmas[45]. However, impact-induced thermal anomalies are absorbed on timescales on the order of one hundred millions of years, even for an event producing a basin the size of the high-Mg region. The high-Mg region encompasses a large range of surface ages[1,15,45], and it is difficult to reconcile a terrain with a large range of ages with a compositional signature characteristic only of the first 100 Myr following the impact[45]. From the point of view of our model, a large impact cannot explain the high-Mg region. This conclusion is consistent with independent analyses based on hydrocode simulations and mineralogical experiments[14,45]. Thus, a crucial question that future thermo-chemical models should investigate in relation to the high-Mg region is the role of heterogeneities in the mantle.

Large impacts modify the source depth of the melt region. This modification varies as a function of the impactor properties, of the timing of the impact event, and of the pattern of the convection in the mantle. Volcanism globally resurfaced Mercury in an interval of time corresponding to the late heavy bombardment[1]. The large amount of impacts that characterize this phase of the history of Mercury and their effects on the magmatic activity in the mantle as inferred here provide an additional mechanism to produce the geochemical anomalies observed today on the surface of the planet.

## Methods

**Convection**. We compute the two-dimensional (2D) convection in the mantle of Mercury using the code GAIA[27]. We use a cylindrical domain where the core and mantle radii are rescaled to reproduce the curvature of the spherical planet[48]. The scaling is obtained by making the ratio of the areas of the cylindrical, 2D domain the same as for the spherical domain. The domain thickness $D$ is set to 419 km, corresponding to the thickness of the silicate shell of Mercury[53]. We employ a half-cylindrical grid with 84 uniformly spaced layers, corresponding to a radial resolution of about 5 km. We use the extended Boussinesq approximation[54] to solve the non-dimensional conservation equations of mass,

momentum, and thermal energy:

$$\nabla \cdot \mathbf{u} = 0, \tag{1}$$

$$-\nabla P + \nabla \cdot \left[ \nu \left( \nabla \mathbf{u} + (\nabla \mathbf{u})^{\mathrm{T}} \right) \right] = Ra\,T\mathbf{e}_r, \tag{2}$$

$$\frac{\partial T}{\partial t} + \mathbf{u} \cdot \nabla T = \nabla \cdot (k\nabla T) + Di\,u_r(T + T_{\mathrm{S}}) + \frac{Di}{Ra}\Phi + \frac{Ra_{\mathrm{Q}}}{Ra}. \tag{3}$$

Velocity, dynamic pressure, temperature, and temperature of the surface are indicated with $\mathbf{u}$, $P$, $T$, and $T_{\mathrm{s}}$, respectively. Symbols $\nu$ and $k$ indicate dynamic viscosity and thermal conductivity, respectively. The radial direction is indicated by $\mathbf{e}_r$ and the radial component of the velocity with $u_r$. Viscous dissipation is indicated with $\Phi \equiv (\boldsymbol{\tau}{:}\boldsymbol{\epsilon})/2$, with $\boldsymbol{\tau}$ the deviatoric stress tensor and $\boldsymbol{\epsilon}$ the strain rate tensor. The dimensionless numbers $Ra$, $Ra_{\mathrm{Q}}$, and $Di$ denote the thermal and internal heating Rayleigh numbers and the dissipation number, and are defined as:

$$Ra = \frac{\alpha g \Delta T D^3 \rho^2 C_{\mathrm{P}}}{\nu_{Ref} k}, \tag{4}$$

$$Ra_{\mathrm{Q}} = \frac{\alpha g D^5 \rho^3 C_{\mathrm{P}} H_r}{\nu_{Ref} k^2}, \tag{5}$$

$$Di = \frac{\alpha g D}{C_{\mathrm{P}}}, \tag{6}$$

where $\rho$ is the mantle density, $C_{\mathrm{P}}$ the heat capacity, $\alpha$ the thermal expansivity, $g$ the acceleration of gravity (which is approximately constant throughout the mantle), and $\Delta T$ the temperature variation across the mantle. The radiogenic heat production in the mantle is indicated by $H_r$. We adopt a 1D parametrized model for the cooling of the core, assumed to have a constant density and heat capacity[55].

The temperature- and pressure-dependent viscosity is calculated with an Arrhenius law for diffusion creep, which in dimensional form reads:

$$\nu(P, T) = \nu_{Ref} \exp\left( \frac{E + PV}{RT} - \frac{E + P_{Ref} V}{RT_{Ref}} \right), \tag{7}$$

where $\nu_{Ref}$, $E$, $V$, and $R$ are the reference values for the viscosity, the activation energy, the activation volume, and the gas constant, respectively. Values for the parameters used in the simulations are listed in Supplementary Table 2.

We model the megaregolith layer as a surficial shell of constant thickness. It affects convection only through its low conductivity. Although this layer would also have low density, we do not consider the density difference since it does not participate in convection and it has a negligible mass, at most 1% of the total silicate mass.

With respect to the models of Tosi et al.[8] we do not employ the particle-in-cell method[56] to calculate crustal production and heat-producing element depletion of the mantle as a consequence of partial melting. However, we reproduced the results of Tosi et al.[8] and Grott et al.[28] by accounting for latent heat and crustal enrichment as described below.

**Latent heat**. Whenever in a grid cell $i$ the local temperature $T_i$ rises above the local solidus temperature $T_{\mathrm{Sol}}$, we equate the "super-solidus" energy

$$E_{\mathrm{S}} = C_{\mathrm{P}} \Delta T, \tag{8}$$

where $\Delta T = T_i - T_{\mathrm{Sol}}$, to the energy required to melt a fraction $\varphi_i$ of the volume of the cell and to compute the variation in temperature $\Delta T_i$ ($< \Delta T$) of the unmolten part $(1 - \varphi_i)$. $\Delta T_i$ represents the local increase in the solidus temperature, which is obtained by solving the equation

$$E_{\mathrm{S}} = L_{\mathrm{m}}\varphi_i + C_{\mathrm{P}} \Delta T_i (1 - \varphi_i), \tag{9}$$

where $L_{\mathrm{m}}$ is the latent heat of melting. We assume instantaneous melt extraction since timescales of melt percolation are faster than convection timescales[57–59]. Partial melts in the Mercurian mantle are buoyant over the entire mantle depth range[33], so we always extract melt to form the crust. The volume of the melt $V_{\mathrm{Melt}}$ produced under a basin of angular extension $\theta$ corresponds to a physical thickness $h = R_{\mathrm{P}} - R_{\mathrm{Melt}}$, where $R_{\mathrm{Melt}}$ is calculated through the expression

$$V_{\mathrm{Melt}} = \frac{\theta}{2} \left[ R^2 - R_{\mathrm{Melt}}^2 \right]. \tag{10}$$

Since we use a 2D grid, $V_{\mathrm{Melt}}$ should be regarded as an area.

**Heat sources**. We allow the HPE to decay according to the standard exponential decay law[60]. Since HPE are highly incompatible, they concentrate in the partial melt. In order to take this process into account, we correct the internal heating

Rayleigh number by a factor $(1 - \Lambda\varphi_t)$ at each time step, where $\Lambda$ is the enrichment factor and $\varphi_t$ is the total melt produced in the mantle in that time step. We consider the same enrichment factor for the three radioactive elements[8,28]. Potassium provides the largest contribution to the heat production in the earliest phases of the evolution, when most of the crust is produced. For each HPE model, we estimate the enrichment factor $\Lambda$ by calculating the ratio between the observed amount of potassium on the surface[61] and the bulk abundance (Supplementary Table 1).

**Comparison with thermo-chemical models.** Tosi et al.[8] showed that 1D parameterized models are in excellent agreement with 2D thermo-chemical models. To validate our thermal convection model, we compared its results on crustal production, core-mantle boundary (CMB) temperature, and stagnant lid thickness with results of 1D parameterized models[28]. In Supplementary Fig. 9, we plot the temporal evolution of the crustal thickness and of the core temperature for a 1D and a 2D simulation. The parameters used are listed in Supplementary Table 4. The agreement is very good, with differences in crustal thickness and CMB temperature <5% throughout the evolution. The 2D model is slightly colder than the 1D case, but in the first billion year of the evolution, when the majority of the crust is produced, the difference is <20 K. A direct comparison with the lid thickness is not meaningful, since Grott et al.[28] use a thermodynamical definition, while we use a more appropriate dynamical definition based on the gradient velocity method[62]. The final value, however, agrees within 5%. We note that the main results of the paper rely on the crustal production curve (Fig. 1 and Supplementary Fig. 2), whose shape (i.e., its three regimes) is similar across the entire range of parameters and is also similar to the crustal production curve obtained with thermo-chemical models (e.g., Fig. 6 in Tosi et al.[8]). As long as the model produces a crustal thickness compatible with the inferred value[10] and predicts melt production for an interval of time compatible with the dating of the most recent large volcanic units[6], the details of the model do not affect our conclusions (compare Fig. 5 and Supplementary Fig. 5).

**Source depth of melt.** At each time step, melt is extracted vertically and contributes to the formation of the crust or the melt sheet. The mean source depth of the melt is computed by weighting the depth of each grid cell with the corresponding melt fraction. The source depth of the melt is deeper for higher reference viscosities, a result consistent with classical boundary layer theory, since the thickness of the boundary layer, and thus the depth of the top of the convecting region, where most melt is produced, is proportional to the reciprocal of the Nusselt number $Nu$ and thus to the reference viscosity through the Rayleigh number $Ra$: $Nu^{-1} \propto Ra^{-\beta} \propto \nu_{Ref}^{\beta}$, where $\beta$ has a value of about 1/3.

**Thermal contraction.** Variations in the radius of Mercury $R_P$ between time $t_i$ and time $t$, induced by variations in the temperature profile, can be calculated from ref. [8]

$$\Delta R_P(t, t_i) = \alpha_c [T_{CMB}(t) - T_{CMB}(t_i)] \frac{R_{CMB}^3}{R_P^2} + \frac{1}{R_P^2} \int_{R_c}^{R_P} \alpha [T_m(r, t) - T_m(t_i)] r^2 \, dr, \tag{11}$$

where the symbols are defined in Supplementary Table 2. In computing the thermal contraction of the baseline model, we consider $t = 4.5$ Gyr and $t_i = 0.75$ Gyr, the latter value corresponding to the peak mantle temperature (Supplementary Fig. 4).

**Volume of the smooth plains and melt production regimes.** The smooth plains represent the youngest large volcanic units of the surface of Mercury. They account globally for about 27% of the surface of Mercury[3]. An upper limit for the volume of the smooth plains is obtained by assuming that they are all volcanic in origin and that they have a thickness of 2 km, as in the case of the northern plains[5]. Assuming that the plains cover a spherical cap with an area $A_{SP}$ corresponding to 27% of the surface, their volume $V_{SP}$ is

$$V_{SP} = A_{SP} \times 2 = 8\pi R_M^2 \times 0.27, \tag{12}$$

which corresponds to about 1.6% of the entire volume of the crust for a crustal thickness of 35 km[10]. This figure increases by an order of magnitude if an extrusive-to-intrusive ratio of 10% is assumed. We use a value of 16% for the total volume of the volcanic plains to define the three melting regimes in the main text.

**Data availability.** The output of the thermal evolution simulations are available upon request from the corresponding author.

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

## Acknowledgements

We thank E. Martellato, K. Miljković, B. Denevi, and M. Massironi for helpful information on surface dating, crater data, and impact simulations. The careful and thoughtful comments of S. Marchi and two anonymous reviewers improved the quality of the paper. Figures where created with the free software described by Ahrens et al.[63] and Hunter[64]. S.P. acknowledges support from the German Academic Exchange Service (DAAD). S.P. and N.T. acknowledge support from the Helmholtz-Gemeinschaft (project VH-NG-1017). A.-C.P. acknowledges support from the Interuniversity Attraction Poles Programme initiated by the Belgian Science Policy Office through the Planet TOPERS alliance and by the Deutsche Forschungsgemeinschaft (SFB-TRR 170). T.R. was supported by the German Research Foundation (DFG, grant Ru 1839/1–1) and by SFB-TRR 170. Computational time has been provided by the HLRN (project bep00041), which is gratefully acknowledged. This is TRR 170 Publication No. 20.

## Author contributions

S.P. conceived the project and wrote the manuscript. S.P. and N.T. performed the numerical simulations. A.-C.P. helped in running the GAIA simulations. T.R. provided input on impact scaling laws. All the authors contributed to the discussion of the results and to the finalization of the manuscript.

## Additional information

**Competing interests:** The authors declare no competing financial interests.

