## [Peer Review File · Nature Communications]

Reviewers' comments:

Reviewer #1 (Remarks to the Author):

This manuscript studies the effects of large collisions on Mercury via new geophysical simulations. This is a very complex problem, and the manuscript relies on a number of assumptions. While, the final outcome may significantly depend on these assumptions (see below), I think there is merit in this approach.

The current text is not easy to follow and could significantly be improved, both in terms of logic flow and content. As such, I do not recommend publication in the present form, but I do think this work can meet Nat Comms standards after a thoroughly revision. Therefore, I look forward to receiving a revised manuscript.

I provide below some specific points to consider:

1. In 7-9: The first two lines do not really convey what is the problem the manuscript is trying to solve (the entire abstract, also, is not in Nature style). For instance, spectral and physical anomalies are mentioned (In 10), but they are not clearly described.
2. In 13-14: Do not use cratering ages with two significant decimal figures. Use here 3.7-3.8 Ga. Furthermore, "corroborated by the dating of the youngest large volcanic units"  This is not necessarily true. The lack of young large volcanic units could also be due to the fact that there are no large young basins capable to trigger volcanism (this is one of the point made by Marchi et al Nature 2013).
3. In 15: "stagnant lid, a volume of the mantle" -> Should this be the "crust"? The overall meaning of the sentence is obscure. In am not sure about the value of the text at In 15-17.

The main conclusion of this work, I think, is that impact-triggered volcanism can significantly postdate an impact. Furthermore, the outcome of a large collisions depends on the pattern of convection at the impact location. I suggest to make these two points the main message of the manuscript.

4. In 18 "comparatively", comparatively to what? Perhaps, use "relatively"?
5. In 24: "diminishing or negligible", I agree this a plausible explanation, but see also comment #2. In addition, secular contraction of the planet likely prevented volcanism to occur at a later time, even if deep-seated melt was still present.
6. In 46: Here are some other potentially relevant papers: Marchi et al PSS 2011; Fegan et al Icarus 2017. I am sure there is more to be quoted here.
7. In 58: It is not obvious why crustal thickness increases with higher HPE concentration, can you comment on that?
8. In 68: I am not an expert about volcanism, but why does the extrusive-to-intrusive ratio matter here? And why the chosen number -65%- is reasonable? (same comment applies elsewhere in the manuscript when the extrusive-to-intrusive ratio is mentioned).
9. In 75/78: What does "last melting event" mean? Isn't melt being continuously produced, even though with a decreasing efficiency over time?
10. In 66-84: This is a rather long description of the three regimes, based on a bunch of

questionable assumptions (e.g., constant 10% extrusion-to-intrusive rate). So, I wonder if there is a more terse way to introduce the three regimes.

11. In 89-90: The scaling law gives shock melting only, so this is worth clarifying.

12. In 98: What is the impactor size used for Caloris? Also in the caption of fig 2 it is stated that a velocity of 15 km/s has been used, but shouldn't have been $43 \cdot \sin(45) \sim 32$ km/s

13. In 104: "basin" -> This is not the correct term, perhaps use "thermal anomaly".

14. In 109-110: "the possible neutral buoyancy horizon" -> What does this mean?

15. In 111: "fractured mantle" -> Not sure it makes sense to have a fractured mantle, perhaps you meant crust?

16. In 112: "upward migration of the melt" -> In addition, the melt has lower density than surrounding mantle, so it is buoyant. See discussion in Marchi et al Nature 2014.

17. In 120/122: What does the term "brief" mean in this context?

18. In 127/129: What do the terms "masked" and "loses any signature" mean in this context?

19. In 130-141: "Based on the large...." This section is speculative and I don't think it is well justified. Crustal thickness in proximity of an impact basin is strongly affected by the dynamic of the impact process (compression, excavation, rebound). This is not accounted for in this model, so I am not sure it is meaningful to have this discussion here. Unless the authors can provide better arguments, I suggest removal.

20. In 143/152: While I agree that the volcanic plains associated with Caloris and Rembrandt could be younger than the basins themselves, it is also worth mentioning that 100-200 Myr is probably close to the analytic error in this kind of estimates (see Marchi et al AJ 2009; PSS 2011).

21. In 173: "source depth of the melt". Notice that the depth of the melt is comparable with the size of the projectiles, so I am not sure how much this estimate is reliable given that the estimate neglects excavation and other dynamic processes associated with a collision. For instance, the analytic expression for the post-impact temperature anomaly does not take into account crater topography.

22. In 186: "The mineralogy of Caloris interior plains is unique", as noted early on, it is never stated what the uniqueness is. What is unique, and why your model can explain that? Need to be more specific here.

23. In 195-200: The Mg-rich region is one the most intriguing observations made by MESSENGER. It is stated here that this terrain is not associated with an ancient impact, but it is not clear why the authors think that is the case. I think it is instead likely that these terrains are associated with a very ancient impact. On Earth, ultramafic rocks, e.g. komatiites, are very rich in Mg. So, I wonder if an early large impact could have produce a massive, deep Mg-rich plume that could have spread over the surface. It should be noted that if the amount of melt far exceeds that of the transient cavity, the melt is likely to spread at the surface erasing the impact signature (Marchi et al Nature 2014).

24. Fig 3: Impact velocity is 43 km/s, but should have been 32 km/s (see comment above). Also the cartoon of the thermal anomaly is really deceiving. It looks like the thermal anomaly is a circle (sphere), but that is not correct. Use the actual scaling law profile.

Table 1: These are parameters for the Earth, why are they also valid for Mercury?

Simone Marchi

Reviewer #2 (Remarks to the Author):

I have reviewed the manuscript "Physical and spectral properties of Mercury's large basins explained by mantle convection and impacts" by Padovan and colleagues, submitted for publication in Nature Communications. The paper describes a set of computational geodynamic models of the evolution of Mercury's mantle with the goal of reproducing the observed volumes of melt sheets in Caloris and Rembrandt basins. The methodology generally appears to be sound, although I have a number of questions that would need to be answered before I could recommend publication. The paper reports an interesting correlation between the age of an impact event and the source depth of the ultimate melt sheet. Being able to differentiate between the direct impact melt and subsequent melting of mantle material enabled by the impact is an important result for Mercury interior studies, although the paper does not go on to discuss the expected mineralogy of melt sheets derived from the different source regions in any detail. Below, please find my specific comments on the manuscript.

1. The title is somewhat misleading, as there is no real analysis of the spectral properties of the basins in the manuscript. The paper hints at the potential for such given that the mineralogy of the different source regions ought to be different, but leaves it to a future study.
2. Lines 33–43. It's not clear that the "three different processes" for inducing melting by an impact are really all different. I think it's actually just two. The unloading of the surface is certainly distinct from the shock heating. But the shock heating is really what interacts with the pre-existing temperature field. So the first and third process are the same thing, just in different regions.
3. Line 50. Are the HPE initially distributed uniformly, or is there some primordial crust that's enriched in them?
4. How is the regolith layer being modeled? Is it just a low-conductivity portion of the lithosphere? Only the conductivity and thickness are given in Table 2. Is it low density as well? Is it permitted to participate in convection? Is it actively prevented from participating in the convection, or are the authors relying on the development of a thick stagnant lid for that? From Figure 4 it looks like that's fine. Ultimately, I don't think it's a big deal, but the details should be given.
5. Line 58. The text states that the crustal thickness increases with the amount of HPE, although presumably there is a limit. If the crust thickens to the point where it crosses the source depth, is it permitted to re-melt? Is this even a concern? What is the source depth as a function of HPE/regolith?
6. Line 95. "five different epochs"  "five different times". How were the times in the waning melting regime chosen?
7. Line 98. What are the chosen impact parameters consistent with a Caloris-sized basin. There could of course be multiple answers. The paper references 45° and 42.5 km/s for the impact angle and velocity, although the 2D model can only handle the vertical component of the velocity (this is ok). And there is quite a distribution in velocity about the mean. The size of the effective heated region scales (inversely) with the velocity and trades off with the intensity of heating within that region, even though all impacts in the suite may deliver the same energy and excavate the same size basin. So some variation in this parameter space should be looked at.

8. Lines 129–141. The text is reasonably clear about how early on, the shallow direct melting is overwhelmed by the subsequent plume melting, and that later, as the convective vigor decreases, the shallow melting dominates. However, I have a hard time seeing these results in the referenced Figure 3, which aggregates all the melt together. Figure 3 shows this schematically at the bottom, but it would be nice to see each component tracked as in the upper plot. Snapshots of this information are shown in Fig. S1, but not the full time evolution.

9. Line 166. What is the solidus being used here?

10. Figure 1. The legend is somewhat confusing. I like how the symbols are allocated for each case, with one property of the symbol corresponding to one model property. I recommend keeping that. But it would be clearer in the legend to show the full symbol used for each case.

11. Figure 3. The caption contains a lot of description about the models that should really go in the main text.

12. Line 203. How are the radii rescaled? I understand some adjustment is needed to reconcile the cylindrical and spherical geometries, but I'm fuzzy about the details.

13. Line 261. Are the comparisons with the 1D parameterized and 2D thermo-chemical convection models shown anywhere? If not in a companion paper, then showing something here would be good.

Reviewer #3 (Remarks to the Author):

The authors present the results of numerical simulations of mantle convection and melting on Mercury that they used to understand the thermal evolution of the mantle and the formation of the secondary volcanic crust. In addition to this, they investigated the effect of large impacts on melt production. They suggest that the mantle had three thermal regimes over time: vigorous melting, waning melting and absent melting. Impacts during the period of vigorous melting will produce only a little amount of melt at shallow depth in the mantle due to the impact. Craters will therefore be filled by lavas originating from deep melting in the mantle due to intense convection. In contrast, impact during the period of absent melting will only produce melting in the shallow mantle and lava sheets will correspond to the melting products of the upper mantle only.

The paper is really interesting and really has the potential to be published in Nature Communications. The text is very easy to read, the arguments are easy to follow and the figures are of extremely high quality. The manuscript however ignores some results from previous studies and additional discussion could therefore be implemented in the manuscript.

Important comments:

1. My first comment is on the results of the thermal model of the mantle. The thermal evolution of the mantle that is presented in Fig. S2 is quite different to that presented in Tosi et al. (2013) although the authors pretend that the results are quite similar. In this new contribution, the temperature of the mantle-core boundary (and the average temperature of the mantle) continuously decreases with time while it increases from 0 to 1500 MYr in Tosi et al. 2013. This difference is really important for the history of mantle melting and crust production, and should be explained.

2. The results presented in Fig. 1 are extremely interesting because they show that most of the volcanic crust is formed after 500 MYr of magmatic activity. I'm however confused by the depth of

the mantle sources. I find the use of a weighted mean source depth a little difficult to understand (it would perhaps be important to show the initial depth of melting) but my conclusion from observing Fig. 1 is that the mantle source doesn't get shallower with time (from 0 to 500 MYr) for any of the viscosity, HPE or regolith conditions used in the models. This is deep contrast with experimental data from Namur et al. (2016; EPSL) and Vander Kaaden and McCubbin (2016; GCA), which show that old lavas are produced at greater depth than young lavas from the Northern Volcanic Plains. How can this be reconciled with the results of your model?

3. In a similar way, phase equilibrium experiments on two compositions from Mercury's crust (High-Mg region and NVP) show that magmas forming the volcanic crust were extracted from the mantle residue at 0.8 GPa (Namur et al., 2016). Considering a mantle depth of 420 km and a mantle-core boundary at 5 GPa, melt extraction from the mantle therefore occurred at ~ 65 km. This is in deep contrast with the modelling results shown in Fig. 4 that suggest that the minimum thickness of the stagnant lid is ~ 85 km. The discrepancy between modelling and experimental results should be better discussed in the paper.

4. The use of Katz's solidus is possibly the major issue of this paper and maybe explains why modelling results do not agree with phase equilibria. First, Katz' model is parameterized for hydrous melting of an Earth's like peridotite. Mercury's mantle is likely to be dry, iron-free, sodium- and sulfur-rich and possibly has a much lower Mg/Si ratio. The solidus is therefore really different than a peridotite on Earth. Some more realistic models could be used. Estimates of a more realistic solidus can be found in Berthet et al. (2009; GCA); Namur et al. (2016) or using experiments in the CMAS system. Correction of the solidus for the absence of FeO could also be done using the recent equations from Kiefer et al. (2015; GCA).

5. Finally, I think that the paper and the discussion section spend quite a long time discussing young craters but do not really discuss old craters. It would perhaps be good to discuss in more details the differences in physical and spectral properties and make a better link between those and the modelling results (impact melting \pm convection melting).

Minor comments:

1. P1; L1-2 – I find the title a bit hard to understand without knowing what physical and spectral properties are considered
2. P1; L10 – Same as in the title; hard to understand what properties are considered
3. P1; L19 – Lavas as old as 4.2 Ga are described in Weider et al. 2012
4. P2; L27-28 – It is not fair to say that the link between melting processes and magma compositions are unknown...this is extensively discussed in Namur et al. 2016 as is the termination of magmatism at 3.5 Ga and the origin of the High-Mg (which is also discussed at the end of this new study)
5. P3; L44-45 – It would be good to give the timescales of the temporal delay between basin formation and volcanic infilling
6. P3; L48 – See my major comment 4 above.
7. P3; L51 – The choice of HPE abundances should be much better explained. There is a justification in the 'method section' but it is not sufficient to understand the values. Values reported in Table S1 come from Padovan et al. 2015. I went back to the original papers that are cited and I understood what the values from Lyubetskaya are. I however do not understand what are the values from McDonough and Sun. They do not correspond to concentrations of CI and even when considering a huge core on Mercury (65 wt.%), I cannot obtain such high values of HPE in the mantle. They also seem much higher than HPE element data from Tosi et al. 2013. Values presented in Table S1 seem to be one order of magnitude too high (at least for U and Th). This must have a strong effect on the thermal models and this should be discussed. In addition, it is commonly assumed that building blocks of Mercury are EH or CB (e.g. Malavergne et al., 2010; Chabot et al., 2014). Why not using HPE values from these meteorites in the thermal models?
8. P3; L54 – This thickness of the regolith should be justified

9. P3; L61 – The effect of the regolith on the mantle melting could be better explained
10. P4; L66-70 – The ratio of extrusive vs intrusive is really important for the estimate of crust production. This ratio should be explained. What are these values based on? Is it comparable to what we see on Earth? How is this influenced by magma density?
11. P5; L92 – Why does Fig. 2 show a simulation with a velocity of 15 km/s? The average velocity of impacts on Mercury is 42.5 km/s (L101). Why not using this value in Fig. 2? That would be better to understand the models shown in Fig. 3 (obtained with a velocity of 42.5 km/s).
12. P6; L109 – In Fig. 3 is there any way to represent the degree of melting associated to impacts or adiabatic decompression?
13. P7; L132-141 – The model described here explains the difference in crustal thickness beneath young and old craters. It however doesn't explain the change in magma composition. Magmas in Caloris are different to those from the surrounding terrains. For Caloris, what degree of melting of the stagnant lid does the model predict? My point is that I do not really understand why the melts produced in the stagnant lid would be compositionally really different to those produced a bit deeper in the convective mantle? Do the authors consider a differentiated magma ocean or a compositionally homogeneous mantle?
14. P8; L157 – how was this value estimated?
15. P9; L170-173 – The thickness of the stagnant lid should be compared with experimental results on Mercury's compositions.

Figure and methods

Fig. 2. Is there any way to show the effect of the age of the impact (t) and the velocity of the impactor on the results?

P18; L25 – The choice of HPE vales should be explained in more details

Table S2 – It would be important to add references for some parameters. e.g. densities, latent heat, ...

Reviewer #1 (Remarks to the Author):

Referee report for "Physical and spectral properties of Mercury's large basins explained by mantle convection and impacts", by Padovan et al.

This manuscript studies the effects of large collisions on Mercury via new geophysical simulations. This is a very complex problem, and the manuscript relies on a number of assumptions. While, the final outcome may significantly depend on these assumptions (see below), I think there is merit in this approach.

The current text is not easy to follow and could significantly be improved, both in terms of logic flow and content. As such, I do not recommend publication in the present form, but I do think this work can meet Nat Comms standards after a thoroughly revision. Therefore, I look forward to receiving a revised manuscript.

Dear Dr. Marchi,

Thank you for your comments, in particular for pointing at the importance of highlighting the results on impact-triggered volcanism. You find below a point-by-point response (line numbers quoted below refer to the revised manuscript).

Sincerely,

Sebastiano Padovan

I provide below some specific points to consider:

1. In 7-9: The first two lines do not really convey what is the problem the manuscript is trying to solve (the entire abstract, also, is not in Nature style). For instance, spectral and physical anomalies are mentioned (In 10), but they are not clearly described.

We agree with with you and we changed the title of the manuscript.

2. In 13-14: Do not use cratering ages with two significant decimal figures. Use here 3.7-3.8 Ga. Furthermore, "corroborated by the dating of the youngest large volcanic units"  This is not necessarily true. The lack of young large volcanic units could also be due to the fact that there are no large young basins capable to trigger volcanism (this is one of the point made by Marchi et al Nature 2013).

We modified the cratering ages as suggested (line 14).

Impacts do facilitate volcanism (a point we make in relation to the extrusive-to-intrusive ratio for Caloris and Rembrandt) but a contracting planet does not necessarily imply that any surface manifestation of volcanic activity is precluded. An order of magnitude estimate of the change in the stress due to cooling and contraction can be obtained as follows: indicating with K the bulk modulus, α the thermal expansivity and DT the variation in temperature, the cooling stress is obtained as $K\alpha DT$. Taking the representative values $K=70$ GPa, $\alpha=3\times 10^{-5}$ K⁻¹, and $DT=100$ K, the cooling stress is about 0.2 GPa. A planet contracting by 7 km (Byrne et al., 2016) corresponds to an additional stress KDR/R , with DR the radial contraction and R the planetary radius. Using 7 km for the radial contraction and 2439 km for the planetary radius, gives a contractional stress similar to the cooling stress of about 0.2 GPa. The cooling and contractional stresses are of the same order as the lithospheric strength (e.g., Mallard et al., Nature (2016)). This estimate provides a straightforward explanation for the appearance

of thrust faults: once contraction and cooling produce a stress comparable to the lithospheric strength, the latter breaks and lobate scarps, manifestation of deep seated thrust faults, appear on the surface. These thrust faults are weakness zones in the lithosphere, and thus provide conduits for magma ascent, if enough magma were produced in the mantle. The setting is similar to Io, a compression-dominated body that is also the most volcanically active body in the solar system (e.g., Bland and McKinnon, Nat. Geosci., 2016). While contraction adds up stress in the lithosphere that would prevent volcanic eruptions, once these stresses are large enough to break the lithosphere, as is the case for Mercury, volcanic material would have conduits to reach the surface even without impacts.

3. In 15: "stagnant lid, a volume of the mantle" -> Should this be the "crust"? The overall meaning of the sentence is obscure. I am not sure about the value of the text at ln 15-17.

The crust is a chemically defined layer, while the stagnant lid is a dynamically defined layer. The volume of the former is, at most, equal to the volume of the latter. If the crust were to become thicker than the lid, its bottom part would be recycled into the mantle. However, due to the rapid development of a thick stagnant lid, in our simulations the crust is always thinner than the stagnant lid. The text provides an indication of the depth and possible composition of source region of the material in the melt sheets. This is valuable information for the interpretation of the spectral data.

The main conclusion of this work, I think, is that impact-triggered volcanism can significantly postdate an impact. Furthermore, the outcome of a large collisions depends on the pattern of convection at the impact location. I suggest to make these two points the main message of the manuscript.

We agree that these two points should be better highlighted. We have therefore rephrased the abstract accordingly and added a dedicated paragraph in the Discussion section.

4. In 18 "comparatively", comparatively to what? Perhaps, use "relatively"?

We modified the text as suggested.

5. In 24: "diminishing or negligible", I agree this a plausible explanation, but see also comment #2. In addition, secular contraction of the planet likely prevented volcanism to occur at a later time, even if deep-seated melt was still present.

Please see response to comment #2.

6. In 46: Here are some other potentially relevant papers: Marchi et al PSS 2011; Fegan et al Icarus 2017. I am sure there is more to be quoted here.

We added the suggested references.

7. In 58: It is not obvious why crustal thickness increases with higher HPE concentration, can you comment on that?

We now include a more extensive discussion of the effect of HPE on crustal thickness (Lines 71-73).

8. In 68: I am not an expert about volcanism, but why does the extrusive-to-intrusive ratio matter here? And why the chosen number -65%- is reasonable? (same comment applies elsewhere in the manuscript when the extrusive-to-intrusive ratio is mentioned).

We use the extrusive-to-intrusive ratio in two different contexts:

First, we assume a value of 10% to provide a conservative estimate of the volume associated with the smooth volcanic plains. This choice is not critical and is used to compute a conservative estimate of the volume of the smooth plains.

Second, for Caloris and Rembrandt, we evaluate which ratio can reconcile the total amount of post-

impact melt produced with the observed volume of the melt sheet. This ratio must be justified given our knowledge of volcanic processes on Earth, for which a value of 20% is usually quoted as representative (White et al., 2006). In the event of a large impact, the removal of the crust and fracturing in the volume around the impact site greatly facilitate eruptions. A ratio larger than 30% for both Caloris and Rembrandt is thus acceptable.

9. In 75/78: What does "last melting event" mean? Isn't melt being continuously produced, even though with a decreasing efficiency over time?

With "last melting event" we indicate the termination of volcanic activity. We clarified this in the text (Lines 119-120).

10. In 66-84: This is a rather long description of the three regimes, based on a bunch of questionable assumptions (e.g., constant 10% extrusion-to-intrusive rate). So, I wonder if there is a more terse way to introduce the three regimes.

We considered moving the definition of the three regimes to the Methods section, but since they represent the base for discussion throughout the paper (Fig 1, bottom, and Figures 4 and 5), we decided to keep it in the main text. We streamlined their description (Lines 109-129).

11. In 89-90: The scaling law gives shock melting only, so this is worth clarifying.

We added a couple of sentence to clarify (Lines 132-137).

12. In 98: What is the impactor size used for Caloris? Also in the caption of fig 2 it is stated that a velocity of 15 km/s has been used, but shouldn't have been $43 \cdot \sin(45) \sim 32$ km/s.

We now always state what the diameter of the impactor is (Line 146, 158, and 160). We also include an "Impactor diameter VS impactor velocity" plot in the Supplementary material (Figure S6). (See also response to your comment #24).

13. In 104: "basin" -> This is not the correct term, perhaps use "thermal anomaly".

The impact largely removes crustal and upper mantle material (see also response to comment #8) in a region that corresponds to the final (i.e., observed) basin. Upward-migrating melt from a volume of the mantle below the final basin will likely erupt and contribute to the final melt sheet. Thus, it is the final size of the basin with respect to the underlying convection pattern that controls the final amount of post-impact melt.

14. In 109-110: "the possible neutral buoyancy horizon" -> What does this mean?

Since the melt is buoyant with respect to the mantle material, it will rise. Under the assumption that crustal material is made of similar material as the buoyant melt, its density will be lower than that of the melt. So rising melt will tend to stall at the base of the crust. This stalling depth represents the melt's neutral buoyancy horizon. Overpressure and other mechanisms (e.g., see the quoted paper by Wiczorek et al. 2001) can lead to eruption of this melt.

15. In 111: "fractured mantle" -> Not sure it makes sense to have a fractured mantle, perhaps you meant crust?

Thanks for noting this error. We meant to use the word "lithosphere". In an impact event both the crust and the mantle (the part which is in the lithosphere) would be fractured. The higher overburden pressure and higher temperature in the mantle will close the cracks faster than in the crust, but this effect will contribute nevertheless to the facilitate post-impact eruptions.

16. In 112: "upward migration of the melt" -> In addition, the melt has lower density than surrounding mantle, so it is buoyant. See discussion in Marchi et al Nature 2014.

We have taken into account melt buoyancy by extracting the melt across the entire mantle, in accordance to the results of Vander Kaaden and McCubbin (2015). We now include this information in the main text (Lines 62-64). Before it was only appearing in the Methods.

17. In 120/122: What does the term "brief" mean in this context?

We mean that it happens right after the impact event. We reworded the text to clarify (Line 190).

18. In 127/129: What do the terms "masked" and "loses any signature" mean in this context?

By "masked" we mean that the volume of convective melt greatly dominates the melt budget in the basin. By "loses any signature" we mean that its source depth is almost identical to the source depth of convective melt. We clarified in the text (Line 198 and 202-203).

19. In 130-141: "Based on the large...." This section is speculative and I don't think it is well justified. Crustal thickness in proximity of an impact basin is strongly affected by the dynamic of the impact process (compression, excavation, rebound). This is not accounted for in this model, so I am not sure it is meaningful to have this discussion here. Unless the authors can provide better arguments, I suggest removal.

It is true that we do not model the initial stages of the basin formation. However, hydrocode simulations clearly show that in forming large basins crustal material is removed (e.g., Frank et al., 2017; Potter and Head, 2015). Thus, it is fair to compare the melt produced after the impact with the currently observed crustal thickness within the basin.

20. In 143/152: While I agree that the volcanic plains associated with Caloris and Rembrandt could be younger than the basins themselves, it is also worth mentioning that 100-200 Myr is probably close to the analytic error in this kind of estimates (see Marchi et al AJ 2009; PSS 2011).

We included this important point (Lines 236-237).

21. In 173: "source depth of the melt". Notice that the depth of the melt is comparable with the size of the projectiles, so I am not sure how much this estimate is reliable given that the estimate neglects excavation and other dynamic processes associated with a collision. For instance, the analytic expression for the post-impact temperature anomaly does not take into account crater topography.

Forming Caloris- or Rembrandt-sized basins requires impacting bodies with diameters on the order of several tens of kilometers. Topography on Mercury is very subdued (a few km at most). While there might be an offset of a couple of kilometers, the method we use is indicative of the source depth of the melt.

22. In 186: "The mineralogy of Caloris interior plains is unique", as noted early on, it is never stated what the uniqueness is. What is unique, and why your model can explain that? Need to be more specific here.

We extended the discussion (Lines 286-313).

23. In 195-200: The Mg-rich region is one of the most intriguing observations made by MESSENGER. It is stated here that this terrain is not associated with an ancient impact, but it is not clear why the authors think that is the case. I think it is instead likely that these terrains are associated with a very ancient impact. On Earth, ultramafic rocks, e.g. komatiites, are very rich in Mg. So, I wonder if an early large impact could have produced a massive, deep Mg-rich plume that could have spread over the surface. It should be noted that if the amount of melt far exceeds that of the transient cavity, the melt is likely to

spread at the surface erasing the impact signature (Marchi et al Nature 2014).

We now extensively discuss the high-Mg region (last paragraph). In addition, we note here that if indeed the volume of melt produced was large enough to spread over the rim to erase the impact signature, then this material would not necessarily have a circular distribution, which was one of the basis for the identification of the high-Mg region as a basin. It is possible, however, that given the central source of melt (a spherical crater) and assuming a homogeneous topography around the newly formed crater, the volcanic material spreads radially uniformly outside of the crater rim. If that is the case, then the size of the high-Mg region does not provide any direct information on the actual location of the original rim, which would have a diameter smaller than the assumed ~3000 km. If significantly smaller, then the argument for a large impact would be weaker.

24. Fig 3: Impact velocity is 43 km/s, but should have been 32 km/s (see comment above). Also the cartoon of the thermal anomaly is really deceiving. It looks like the thermal anomaly is a circle (sphere), but that is not correct. Use the actual scaling law profile.

We always assume impacts at 45 degree inclination. The velocity quoted in the text is the encounter velocity not corrected for the inclination. We now state this information in the text (Lines 144-145). The cartoon is just illustrative. The actual thermal anomaly can be seen in Figure 3.

Table 1: These are parameters for the Earth, why are they also valid for Mercury?

Parameters that are not planet-specific refer to commonly adopted values in convection simulations of Mercury. We now include explicit references for the values quoted in Supplementary Table 2.

Simone Marchi

Reviewer #2 (Remarks to the Author):

I have reviewed the manuscript “Physical and spectral properties of Mercury's large basins explained by mantle convection and impacts” by Padovan and colleagues, submitted for publication in Nature Communications. The paper describes a set of computational geodynamic models of the evolution of Mercury's mantle with the goal of reproducing the observed volumes of melt sheets in Caloris and Rembrandt basins. The methodology generally appears to be sound, although I have a number of questions that would need to be answered before I could recommend publication. The paper reports an interesting correlation between the age of an impact event and the source depth of the ultimate melt sheet. Being able to differentiate between the direct impact melt and subsequent melting of mantle material enabled by the impact is an important result for Mercury interior studies, although the paper does not go on to discuss the expected mineralogy of melt sheets derived from the different source regions in any detail. Below, please find my specific comments on the manuscript.

Dear Reviewer,

Thank you for your comments. We followed them, and in the revised manuscript we try to better illustrate the model and we now include results for a slow impactor, which is compatible with the expected impactor velocities at Mercury. You find a point-by-point response below (line numbers quoted below refer to the revised manuscript).

Sincerely,

Sebastiano Padovan

1. The title is somewhat misleading, as there is no real analysis of the spectral properties of the basins in the manuscript. The paper hints at the potential for such given that the mineralogy of the different source regions ought to be different, but leaves it to a future study.

We agree and we changed the title of the manuscript accordingly.

Lines 33–43. It's not clear that the "three different processes" for inducing melting by an impact are really all different. I think it's actually just two. The unloading of the surface is certainly distinct from the shock heating. But the shock heating is really what interacts with the pre-existing temperature field. So the first and third process are the same thing, just in different regions.

We agree with the reviewer that the shock heating generates both the shock melt and, by interaction with the pre-existing temperature field, the post-impact melting which is the focus of this work.

However, they are distinct in their timescales and for this reason we keep them separate.

3. Line 50. Are the HPE initially distributed uniformly, or is there some primordial crust that's enriched in them?

We do not include any primordial crust. The HPE are initially distributed uniformly. We clarified this point in the revised text (Lines 58-59).

4. How is the regolith layer being modeled? Is it just a low-conductivity portion of the lithosphere? Only the conductivity and thickness are given in Table 2. Is it low density as well? Is it permitted to participate in convection? Is it actively prevented from participating in the convection, or are the authors relying on the development of a thick stagnant lid for that? From Figure 4 it looks like that's fine. Ultimately, I don't think it's a big deal, but the details should be given.

The regolith is modeled as a shell of given thickness encircling the planet. Its only influence on the

convection is due to its low-conductivity. Despite such a layer should have low density, we do not consider the density difference since, as the reviewer correctly notes, it does not participate in convection (and its total mass is at most about 1% of the total silicate mass). We now added this information in the Methods section (Lines 361-364).

5. Line 58. The text states that the crustal thickness increases with the amount of HPE, although presumably there is a limit. If the crust thickens to the point where it crosses the source depth, is it permitted to re-melt? Is this even a concern? What is the source depth as a function of HPE/regolith? The reviewer is correct in noting that the bottom part of a thick crust might remelt if it is deeper than the lid at a given time. However, the development of a thick stagnant lid is rapid, and the bottom of the crust is always well above the melt region in the mantle. So the potential remelting of the crust is not a concern for these models of Mercury (Figure 2 now plots the source depth of the melt). The variation of the source depth as a function of the HPE/regolith can be seen from Figure 1 and Supplementary Figure S1. This topic is discussed in the section “Results/Thermal evolution models” (Lines 71-73).

Line 95. "five different epochs"  "five different times". How were the times in the waning melting regime chosen?

The wording of “epochs” has been corrected.

The times are chosen to properly sample the different regimes. We included the 750 Myr on purpose, since this epoch roughly corresponds to the formation of the Caloris and Rembrandt basins.

7. Line 98. What are the chosen impact parameters consistent with a Caloris-sized basin. There could of course be multiple answers. The paper references 45° and 42.5 km/s for the impact angle and velocity, although the 2D model can only handle the vertical component of the velocity (this is ok). And there is quite a distribution in velocity about the mean. The size of the effective heated region scales (inversely) with the velocity and trades off with the intensity of heating within that region, even though all impacts in the suite may deliver the same energy and excavate the same size basin. So some variation in this parameter space should be looked at.

Impactor velocities for Mercury have a wide distribution (LeFeuvre and Wieczorek, 2008). While the mean value corresponds to 42.5 km/s, which we employed in the simulation included in the manuscript, there is a large scatter around it. We added a sentence that describes this information (Lines 142-143). We tested the effect of using a slow impactor (15 km/s) to form Caloris. In this scenario both the amount and timing of post-impact melt vary, but the variations are minor and do not modify the results. We quote the values for this slow-impact case in Lines 242-247.

8. Lines 129–141. The text is reasonably clear about how early on, the shallow direct melting is overwhelmed by the subsequent plume melting, and that later, as the convective vigor decreases, the shallow melting dominates. However, I have a hard time seeing these results in the referenced Figure 3, which aggregates all the melt together. Figure 3 shows this schematically at the bottom, but it would be nice to see each component tracked as in the upper plot. Snapshots of this information are shown in Fig. S1, but not the full time evolution.

Following your comment we modified Figure 2 (which is now Figure 3) and plotted, along with the temperature and velocity fields, the melting field.

9. Line 166. What is the solidus being used here?

For the solidus we use the anhydrous parameterization of Katz et al., 2003. This information is now included in the main text (Lines 64-67). In the revised text, we also use a CMAS solidus, following a suggestion of Reviewer #3. We added a Table (Supplementary Table 3) which lists the parameters we

used for the solidus and liquidus curves and a plot of these curves (Supplementary Figure S1).

10. Figure 1. The legend is somewhat confusing. I like how the symbols are allocated for each case, with one property of the symbol corresponding to one model property. I recommend keeping that. But it would be clearer in the legend to show the full symbol used for each case.

We think that adding the symbols for each single model, along with their properties, would make it cluttered. We prefer to keep it the way it is now. However, to improve clarity, we now use thicker lines for models compatible with the volume of the crust and timing of the volcanic activity.

11. Figure 3. The caption contains a lot of description about the models that should really go in the main text.

We agree, and we extended the discussion in the text. We also kept the original caption since we would like the main results of the paper to be clear by just inspecting the figures and relative captions (this approach is also encouraged by the journal's submission guide).

12. Line 203. How are the radii rescaled? I understand some adjustment is needed to reconcile the cylindrical and spherical geometries, but I'm fuzzy about the details.

We included a sentence explaining that the rescaling is based on making the ratio of the areas of the cylindrical 2D domain the same as for the spherical domain (Lines 332-333).

13. Line 261. Are the comparisons with the 1D parameterized and 2D thermo-chemical convection models shown anywhere? If not in a companion paper, then showing something here would be good.

We added an additional figure (Supplementary Figure S9) that illustrates an example of a comparison between the 1D parameterized and our thermal model for a set of input parameters which are listed in a new Table (Supplementary Table 4). We also extended the discussion in the Methods (Lines 382-409).

Reviewer #3 (Remarks to the Author):

The authors present the results of numerical simulations of mantle convection and melting on Mercury that they used to understand the thermal evolution of the mantle and the formation of the secondary volcanic crust. In addition to this, they investigated the effect of large impacts on melt production. They suggest that the mantle had three thermal regimes over time: vigorous melting, waning melting and absent melting. Impacts during the period of vigorous melting will produce only a little amount of melt at shallow depth in the mantle due to the impact. Craters will therefore be filled by lavas originating from deep melting in the mantle due to intense convection. In contrast, impact during the period of absent melting will only produce melting in the shallow mantle and lava sheets will correspond to the melting products of the upper mantle only.

The paper is really interesting and really has the potential to be published in Nature Communications. The text is very easy to read, the arguments are easy to follow and the figures are of extremely high quality. The manuscript however ignores some results from previous studies and additional discussion could therefore be implemented in the manuscript.

Dear Reviewer,

Thank you for your comments. Most notably, following your suggestion about the use of a different model for the heat producing element, we ran a new set of simulations and we picked a new baseline model, which is compatible to the results of Namur et al. (2016). You find a point-by-point response below (line numbers quoted below refer to the revised manuscript).

Sincerely,

Sebastiano Padovan

Important comments:

1. My first comment is on the results of the thermal model of the mantle. The thermal evolution of the mantle that is presented in Fig. S2 is quite different to that presented in Tosi et al. (2013) although the authors pretend that the results are quite similar. In this new contribution, the temperature of the mantle-core boundary (and the average temperature of the mantle) continuously decreases with time while it increases from 0 to 1500 MYr in Tosi et al. 2013. This difference is really important for the history of mantle melting and crust production, and should be explained.

The original text was not clear in stating that our reference model was not to be directly compared with Tosi's reference model since the latter has a different reference viscosity (4.46×10^{21} Pa s, compared to 10^{20} Pa s), a slightly different mantle thickness (400 km compared to 419 km), and uses a different

solidus (Takahashi 1990, compared to the anhydrous parameterization of Katz et al., 2003).

We now include an additional figure (Supplementary Figure S9) where we compare the results from a run with the same parameters as ours and a corresponding Table with the parameters for the 1D model (Supplementary Table 4).

2. The results presented in Fig. 1 are extremely interesting because they show that most of the volcanic crust is formed after 500 MYr of magmatic activity. I'm however confused by the depth of the mantle sources. I find the use of a weighted mean source depth a little difficult to understand (it would perhaps be important to show the initial depth of melting) but my conclusion from observing Fig. 1 is that the mantle source doesn't get shallower with time (from 0 to 500 MYr) for any of the viscosity, HPE or regolith conditions used in the models. This is deep contrast with experimental data from Namur et al. (2016; EPSL) and Vander Kaaden and McCubbin (2016; GCA), which show that old lavas are produced at greater depth than young lavas from the Northern Volcanic Plains. How can this be reconciled with the results of your model?

Thanks to your comment we found a bug in the way we color coded symbols in Figure 1. While the source depth was indeed getting shallower with time (as correctly shown in the bottom of Figure S2 of the manuscript initially submitted), this tendency was not reflected in the colors of Figure 1. The revised Figure 1 now shows the correct source depth (which does indeed vary, albeit not much). Namur et al. (2016) infer that the melting process in the mantle of Mercury is "batch melting", and their inferred depth refers to the depth of the extraction of the melt from the location of the ancient lithosphere. Therefore, the values reported in Namur (their Fig. 7) refer to the location of the lithosphere. We now have a new Figure 2, where we plot both the location of the stagnant lid and the depth of the source region for three models satisfying both the crustal thickness constraint and the timing of the volcanic activity constraint. Our new baseline model (which is found by following your comment #3 and #4, see below) has a lid thickness that is within the intervals derived in Namur et al. (2016).

3. In a similar way, phase equilibrium experiments on two compositions from Mercury's crust (High-Mg region and NVP) show that magmas forming the volcanic crust were extracted from the mantle residue at 0.8 GPa (Namur et al., 2016). Considering a mantle depth of 420 km and a mantle-core boundary at 5 GPa, melt extraction from the mantle therefore occurred at ~ 65 km. This is in deep contrast with the modelling results shown in Fig. 4 that suggest that the minimum thickness of the stagnant lid is ~ 85 km. The discrepancy between modelling and experimental results should be better discussed in the paper.

In addition to the response to comment #2, we add that the results of Namur present a range of T-P values for each geochemical terrane which correspond to S-devoid and S-rich composition. We use their results, shown in the new Figure 2 (white boxes), to guide the choice of the baseline model.

4. The use of Katz's solidus is possibly the major issue of this paper and maybe explains why modelling results do not agree with phase equilibria. First, Katz' model is parameterized for hydrous melting of an Earth's like peridotite. Mercury's mantle is likely to be dry, iron-free, sodium- and sulfur-rich and possibly has a much lower Mg/Si ratio. The solidus is therefore really different than a peridotite on Earth. Some more realistic models could be used. Estimates of a more realistic solidus can be found in Berthet et al. (2009; GCA); Namur et al. (2016) or using experiments in the CMAS system. Correction of the solidus for the absence of FeO could also be done using the recent equations from Kiefer et al. (2015; GCA).

We use the anhydrous parameterization of Katz (his eq. 4 and 10, with parameters from his Table 2). However, following the reviewers' suggestion, we now include a set of models with a different HPE model, based on enstatite chondrites, and the iron-free, Na-corrected solidus of the CMAS system used

by Namur et al. (2016). This model satisfies all the constraints that we considered (crustal thickness, timing of the volcanic activity) and also the time-evolution of its lid thickness is bracketed by the experimental results of Namur et al. (2016).

5. Finally, I think that the paper and the discussion section spend quite a long time discussing young craters but do not really discuss old craters. It would perhaps be good to discuss in more details the differences in physical and spectral properties and make a better link between those and the modelling results (impact melting \pm convection melting).

A more extensive discussion of the high-Mg region, a potential site of a large ancient impact, is now included in the discussion session (last paragraph).

Minor comments:

1. P1; L1-2 – I find the title a bit hard to understand without knowing what physical and spectral properties are considered

Following your comment we modified the title to better reflect the paper's results.

2. P1; L10 – Same as in the title; hard to understand what properties are considered

We reworded the abstract.

3. P1; L19 – Lavas as old as 4.2 Ga are described in Weider et al. 2012

The ages reported in Table 3 of Weider et al. 2012 are taken from Strom and Neukum 1988. The two references we cite (Marchi et al. 2013 and Whitten et al. 2014) took advantage of the extensive data provided by the MESSENGER mission. Please note, however, that the age of the oldest units on the surface is not important for the conclusions of the paper.

4. P2; L27-28 – It is not fair to say that the link between melting processes and magma compositions are unknown...this is extensively discussed in Namur et al. 2016 as is the termination of magmatism at 3.5 Ga and the origin of the High-Mg (which is also discussed at the end of this new study)

Our “not fully clarified” was referred to the inability of geodynamical models to reproduce/fit the inference based on observations of the surface and/or melting experiments. We removed the references in the sentence containing the segment “not fully clarified”, and left them only in reference to the single processes discussed. References to the work of Namur (Namur et al., 2016; Namur and Charlier, 2017) have been added.

5. P3; L44-45 – It would be good to give the timescales of the temporal delay between basin formation and volcanic infilling

We quote the temporal delay when we compare our model results with the observations in lines 216 for Caloris and 225 for Rembrandt.

6. P3; L48 – See my major comment 4 above.

Please see response to major comment 4 above.

7. P3; L51 – The choice of HPE abundances should be much better explained. There is a justification in the ‘method section’ but it is not sufficient to understand the values. Values reported in Table S1 come from Padovan et al. 2015. I went back to the original papers that are cited and I understood what the values from Lyubetskaya are. I however do not understand what are the values from McDonough and Sun. They do not correspond to concentrations of CI and even when considering a huge core on

Mercury (65 wt.%), I cannot obtain such high values of HPE in the mantle. They also seem much higher than HPE element data from Tosi et al. 2013. Values presented in Table S1 seem to be one order of magnitude too high (at least for U and Th). This must have a strong effect on the thermal models and this should be discussed. In addition, it is commonly assumed that building blocks of Mercury are EH or CB (e.g. Malavergne et al., 2010; Chabot et al., 2014). Why not using HPE values from these meteorites in the thermal models?

Thank you for noting the error in the HPE abundances. In Supplementary Table 1 of the initial submission the abundances for U and Th were indeed off by an order of magnitude, a typo carried over from the Table appearing in Padovan et al., 2015. However, the code has the correct amounts so the results are correct. As for the choice of the HPE model, we followed your suggestion and included an HPE model based on the abundances of enstatite chondrite reported in Wasson and Kallemeyn (1988). This model, for which we use the CMAS solidus of Namur et al. (2016), is now our new baseline model.

8. P3; L54 – This thickness of the regolith should be justified

We added two references to justify the choice of the regolith layer thickness (Line 50).

9. P3; L61 – The effect of the regolith on the mantle melting could be better explained

Additional explanations have been included regarding the effects of the regolith on melting (Lines 73-75).

10. P4; L66-70 – The ratio of extrusive vs intrusive is really important for the estimate of crust production. This ratio should be explained. What are these values based on? Is it comparable to what we see on Earth? How is this influenced by magma density?

We added the sentence “Partial melts in Mercury’s mantle are buoyant over the entire mantle depth range, and they always contribute to the building of the crust” (Lines 62-64). All the melt produced by mantle convection contributes to the crustal production. Depending on the density contrast between crustal material and upward-migrating melt, the latter may stall at the base of the crust. However, it still contributes to crust formation, even for a 100% intrusive rate. Our assumed extrusive-to-intrusive ratio affects only the definition of the three regimes, not the crustal production. The assumed value of 10% is a conservative assumption guided by data we have for Earth (where a value of 20% is taken as reference, White et al., 2006).

11. P5; L92 – Why does Fig. 2 show a simulation with a velocity of 15 km/s? The average velocity of impacts on Mercury is 42.5 km/s (L101). Why not using this value in Fig. 2? That would be better to understand the models shown in Fig. 3 (obtained with a velocity of 42.5 km/s).

Slower impactors generate larger thermal anomalies (Roberts and Barnouin, 2012) and we used a slow impactor in Figure 2 of the initial submission to better illustrate the interaction with the temperature field. However, we realized, thanks to your comment, that using a slow impactor might confuse the reader. Accordingly, we now employ an average speed (42.5 km/s) impactor.

12. P6; L109 – In Fig. 3 is there any way to represent the degree of melting associated to impacts or adiabatic decompression?

A physically meaningful calculation of the degree of melting can only be obtained by modeling a number of processes—two phase flow, stability of liquid and solid phases, etc.--that currently no global geodynamical code is able to perform. This consideration is now included in the revised text (Lines 297-302).

13. P7; L132-141 – The model described here explains the difference in crustal thickness beneath

young and old craters. It however doesn't explain the change in magma composition. Magmas in Caloris are different to those from the surrounding terrains. For Caloris, what degree of melting of the stagnant lid does the model predict? My point is that I do not really understand why the melts produced in the stagnant lid would be compositionally really different to those produced a bit deeper in the convective mantle? Do the authors consider a differentiated magma ocean or a compositionally homogeneous mantle?

We extended the discussion on the potential compositional signatures of melt sheets for ancient and old craters in the discussion session. We point to the necessity, for future studies, to investigate a chemically heterogeneous mantle (our model employs a homogeneous mantle).

14. P8; L157 – how was this value estimated?

We performed simulations similar to those used for Figure 3 of the initial submission, but with parameters for the impactor appropriate for Rembrandt. The figure is now included in the manuscript (new Figure 4).

15. P9; L170-173 – The thickness of the stagnant lid should be compared with experimental results on Mercury's compositions.

We now include the results of Namur et al. (2016) in Figure 2.

Figure and methods

Fig. 2. Is there any way to show the effect of the age of the impact (t) and the velocity of the impactor on the results?

Not that we can think of. However, the most important outcome, volume and source depth of the post-impact melt, are captured by Figures 4 and 5. We also performed simulation using a slow impactor (15 km/s). The relevant numbers are now included in the text (Lines 242-247).

P18; L25 – The choice of HPE vales should be explained in more details

We extended the discussion of the HPE models in the main text (Lines 54-58) and modified the Table listing the abundances (Supplementary Table 1).

Table S2 – It would be important to add references for some parameters. e.g. densities, latent heat, ...

We included the references for the values adopted Supplementary Table 2).

Reviewers' comments:

Reviewer #1 (Remarks to the Author):

I have read the revised manuscript, and I find it much improved. There are still a few issues that need to be addressed, in my opinion, before the manuscript can be published. See detailed comments below.

1. The title does not really say what the main finding of the manuscript is. It is a generic statement. What is it, specifically, that the impact-related volcanism say about the mantle?

2. In 12-15: There is a subtle aspect here. This text implies the drop in impact-related volcanism at about 3.7-3.8 Ga is the result of a weak internal convection. Could also this be due to the lack of large impacts after that time?

3. In 17: "and the solidified impact-induced melt pond", it is not clear what the meaning of this statement is in relation to the previous clause.

4. In 29-31: This text is not very clear. Please rephrase. For instance, how a rapidly cooling mantle could produce a geochemically heterogeneous crust?

5. In 55-56: Regarding the choice of EH- and CI-HPE, it is worth noting that the bulk silicate Mercury is likely very different to both EH- and CI-HPE. Even if bulk Mercury started as EH or CI, core formation and silicate stripping by the giant collision responsible for the high core/mantle ratio could have resulted in a very different mantle composition. In this regard, it is perhaps surprising that the BSE is a poorer model than EH and CI, as stated in the manuscript.

6. In 73/99: Is "regolith" the correct term here? Regolith usually is a very thin surface layer of fine dust. The supplementary material explains what it is meant by regolith, simply and insulating layer. However, regolith is a loaded term, which may have a different flavor that required here.

7. In 101-102: How sensitive is the baseline model contraction of 9.9 km to model parameters? Is any of your other models producing a contraction close to ~ 7 km (as measured)?

8. In 132-137: The new text is not very clear. I think it simply needs to say that shock temperatures are above the solidus, but very quickly drop to the solidus. Thus, your model truncates the impact temperature at the solidus.

9. In 137-141: "Comparison are unimportant". This text can be removed, or moved to the Methods.

10. In 150-151: This text repeats In 147-148.

11. In 194-214: I don't understand the implication for crustal thinning. There are many variables that affect the production of melt in addition to the timing of collision, such as impactor velocity and size. So, I am not sure it is so straightforward to compare inferred crustal thickness to models. I think that this section is rather weak, please consider removal. The manuscript is perhaps too long (also due to numerous repetitions as noted), and this is a good place to cut.

12. In 237-239: The text about extrusive-intrusive ratio sort of repeats text in the previous paragraph.

13. In 243: 15 km/s is very unlikely (very lower limit of the

distribution). I am not sure it is worth describing this case, but I'll leave the authors decide on this.

14. In 248: "Discussion", the new text at the beginning of the section repeats concepts already discussed. I find there are lots of repetitions throughout the manuscript, making the text a bit heavy to read.

15. In 311-313: Remove, it does not add anything substantial.

----Additional note:

16. An important point to stress, in my opinion, is that the depth of impact-induced melt extraction considerably varies as a function of impact timing, impactor size, convection pattern. I would expect this could result in a compositionally heterogeneous surface, even if you have excluded this could explain the high-Mg terrains.

----Rebuttal to my v1 comments:

17. #13: I still think "basin" is not the correct term. You are discussing the deep, internal effects of collisions, which are thus directly connected to the size of the thermal anomaly. The surface basin size has no connection to the internal effects, and merely is the result of excavation and surface collapse, whose properties are not connected to the internal evolution. Also crater modification stage is important in controlling the final basin size, as well as any post-formation degradation.

It is a minor point, but I find it illogical to refer to "basin size" when discussing internal effects.

----Figures:

18. Fig 2: I find this figure confusing. It is supposed to show the "stagnant lid thickness", but the y-axis reports "Depth", this is the same melt source depth used in Fig 1 based on the background color bars, why? I suggest to change y-axis label and to remove the overly confusing background colors.

19. Fig 3: Nice graphic, but to increase clarity please reduce the numbers of panels (eg -1, 0, 22 or 68). Indicate in the caption this is for an impact angle at 45 deg (same for Fig 4 and subsequent figures in which velocity is reported).

20. Fig 6: This figure is a bit confusing. Perhaps consider breaking it into two panels.

Simone Marchi

Reviewer #3 (Remarks to the Author):

Thank you to the authors for the substantial thought put into revising with large-scale implications in mind. I am satisfied with all of the authors responses to my original comments. The responses are thoughtful and were well executed in the manuscript. It seems that the authors also responded carefully to the comments of the two other reviewers.

I am particularly happy to see that the authors now consider a more realistic solidus curve in their calculations and that their original conclusions still hold.

I have no issues with the manuscript as it is. I think it is a substantial contribution to our

understanding of Mercury's volcanic history and the formation of the crust.

My only comment is still about the title. It could still be improved. I'm still not sure that it is easy to understand what the authors mean by 'indicative of thermal state and composition'.

Reviewer #1 (Remarks to the Author):

I have read the revised manuscript, and I find it much improved. There are still a few issues that need to be addressed, in my opinion, before the manuscript can be published. See detailed comments below.

Dear Dr. Marchi,

thank you for your valuable comments, which we addressed in the re-revised manuscript. You find a point-by-point response below (line numbers refer to the re-revised text).

Sincerely,

Sebastiano Padovan

1. The title does not really say what the main finding of the manuscript is. It is a generic statement. What is it, specifically, that the impact-related volcanism say about the mantle?

We changed the title, which now should better reflect the main findings of the paper.

2. In 12-15: There is a subtle aspect here. This text implies the drop in impact-related volcanism at about 3.7-3.8 Ga is the result of a weak internal convection. Could also this be due to the lack of large impacts after that time?

We interpret the age of the youngest large volcanic provinces (YLVP) as indicating that magmatic activity in the interior was diminishing/minor at the time of their emplacement. However, it has been argued (e.g., Byrne et al., 2016) that the compressive state of the lithosphere induced by planetary contraction would prevent further eruptions. If this argument were correct, then the age of the YLVP would not have a causal connection to magmatic activity in the interior. However, the contraction of the planet is due to the cooling of the interior, which in turn indicates decreasing temperatures, and decreasing temperatures correspond to decreasing rates of melt production.

We understand that this might represent a critical point, so we added an explanation in the manuscript (lines 24-30).

3. In 17: "and the solidified impact-induced melt pond", it is not clear what the meaning of this statement is in relation to the previous clause.

We agree, and we reworded the final sentence of the abstract.

4. In 29-31: This text is not very clear. Please rephrase. For instance, how a rapidly cooling mantle could produce a geochemically heterogeneous crust?

We extended the text, which now provides a basic explanation for each of the listed processes.

5. In 55-56: Regarding the choice of EH- and CI-HPE, it is worth noting that the bulk silicate Mercury is likely very different to both EH- and CI-HPE. Even if bulk Mercury started as EH or CI, core formation and silicate stripping by the giant collision responsible for the high core/mantle ratio could have resulted in a very different mantle composition. In this regard, it is perhaps surprising that the BSE is a poorer model than EH and CI, as stated in the manuscript.

We added a sentence (lines 66-68) clarifying that the models are not intended to represent the actual Mercury's composition.

6. In 73/99: Is "regolith" the correct term here? Regolith usually is a very thin surface layer of fine dust.

The supplementary material explains what it is meant by regolith, simply and insulating layer. However, regolith is a loaded term, which may have a different flavor that required here. We substituted regolith with megaregolith everywhere in the manuscript.

7. In 101-102: How sensitive is the baseline model contraction of 9.9 km to model parameters? Is any of your other models producing a contraction close to ~ 7 km (as measured)?

Among the three models considered in Figure 2, the baseline model (white line) has the largest amount of contraction (9.9 km). To first order, the total amount of contraction is proportional to the total amount of cooling between the peak mantle temperature and today. The top-right panel of Figure S4 shows the evolution of the mantle temperature for the models shown in Figure 2. The blue and green models have contractions of 8.6 km and 7.5 km, respectively. While these values are closer to the measured 7 km, the corresponding lid thickness evolution does not match the inferred depth (white boxes in Figure 2). Both the contraction and the location of the lid thickness represent constraints on the thermal evolution of Mercury, and one may argue about which the most stringent one is. We decided to focus on the lid constraint, but we note that our conclusions are unaffected by the choice of the baseline model (see Figure S5).

We also added a few lines (114-116) describing recent results showing that part of the initial radius change is not appearing in the geologic record, possibly adding up to 2.5 km to the observed 7 km (Klimczak, JGR, 2015).

8. In 132-137: The new text is not very clear. I think it simply needs to say that shock temperatures are above the solidus, but very quickly drop to the solidus. Thus, your model truncates the impact temperature at the solidus.

We agree and we shortened the text.

9. In 137-141: "Comparison are unimportant". This text can be removed, or moved to the Methods.

We removed the text.

10. In 150-151: This text repeats In 147-148.

We removed/reworded text to avoid the repetition (lines 154-160)

11. In 194-214: I don't understand the implication for crustal thinning. There are many variables that affect the production of melt in addition to the timing of collision, such as impactor velocity and size. So, I am not sure it is so straightforward to compare inferred crustal thickness to models. I think that this section is rather weak, please consider removal. The manuscript is perhaps too long (also due to numerous repetitions as noted), and this is a good place to cut.

The velocity and size of the impactor do affect the total amount of melt produced, as we illustrate by considering the case of a slow Caloris-forming impact event (see also response to comment #13 below). A Caloris formed with a 42.5 km/s impactor corresponds to ~3.6 km of melt, while a 15 km/s impactor generates ~4.5 km. Thus, while the velocity-size component modifies the resulting thickness of the melt sheet only by a factor of about 1.3, the effect of the timing of the impact produces variations of about one order of magnitude (compare events at $t=300$ Myr and $t=750$ Myr in Figures 4 and 5). Of course we cannot predict accurately the thickness of the crust, but we can draw the first-order conclusion that younger basins should be associated with thinned crust, which is consistent with what crustal thickness maps show.

12. In 237-239: The text about extrusive-intrusive ratio sort of repeats text in the previous paragraph.

We reworded the first part of the paragraph (lines 241-245).

13. In 243: 15 km/s is very unlikely (very lower limit of the distribution). I am not sure it is worth describing this case, but I'll leave the authors decide on this. We include this case because the collision probability for Mercury shows a sort of bimodal distribution with a relatively dense component near small velocities (Figure 5 of Le Feuvre and Wieczorek, 2008). By computing this end-member case we can also estimate the effect of the velocity/size of the impactor on the calculated melt sheet thickness (see also response to comment #11).

14. In 248: "Discussion", the new text at the beginning of the section repeats concepts already discussed. I find there are lots of repetitions throughout the manuscript, making the text a bit heavy to read.

We agree, and we entirely removed the first paragraph of the "Discussion".

15. In 311-313: Remove, it does not add anything substantial.

In terms of the results it does not add anything, but it is important to point to avenues where BepiColombo could provide critical input following the completion of the MESSENGER mission.

----Additional note:

16. An important point to stress, in my opinion, is that the depth of impact-induced melt extraction considerably varies as a function of impact timing, impactor size, convection pattern. I would expect this could result in a compositionally heterogeneous surface, even if you have excluded this could explain the high-Mg terrains.

Thanks for this additional note. We added a concluding paragraph that indicates the possible impact-related origin of at least some of the geochemically varied surface.

----Rebuttal to my v1 comments:

17. #13: I still think "basin" is not the correct term. You are discussing the deep, internal effects of collisions, which are thus directly connected to the size of the thermal anomaly. The surface basin size has no connection to the internal effects, and merely is the result of excavation and surface collapse, whose properties are not connected to the internal evolution. Also crater modification stage is important in controlling the final basin size, as well as any post-formation degradation.

It is a minor point, but I find it illogical to refer to "basin size" when discussing internal effects.

We agree that it is the size of the thermal anomaly that controls the interaction of the energy of the impact with the internal convective processes. However, we try to match the observed melt sheets, which cover the entire basins, therefore we keep track of the melt produced below the final basin.

----Figures:

18. Fig 2: I find this figure confusing. It is supposed to show the "stagnant lid thickness", but the y-axis reports "Depth", this is the same melt source depth used in Fig 1 based on the background color bars, why? I suggest to change y-axis label and to remove the overly confusing background colors.

We reworded the caption to be consistent with the axis labels. We keep the background colors since they provide a direct way of comparing the source depth of the melt sheet in Caloris and Rembrandt (Figures 4 and 5) with the location of the stagnant lid.

19. Fig 3: Nice graphic, but to increase clarity please reduce the numbers of panels (eg -1, 0, 22 or 68). Indicate in the caption this is for an impact angle at 45 deg (same for Fig 4 and subsequent figures in which velocity is reported).

We inserted the information about the impact angle where needed. As long as the format of the figure is consistent with NatComm requirements we keep the entire sequence since it illustrates how the thermal anomaly first focuses nearby melting zones ($t = 4$ Myr) towards the impact region producing a new melting region ($t = 22$ Myr), which persists for tens of Myr ($t = 68$ Myr).

20. Fig 6: This figure is a bit confusing. Perhaps consider breaking it into two panels.
We agree, and we have split the figure in two panels (and modified the caption accordingly).

Simone Marchi

Reviewer #3 (Remarks to the Author):

Thank you to the authors for the substantial thought put into revising with large-scale implications in mind. I am satisfied with all of the authors' responses to my original comments. The responses are thoughtful and were well executed in the manuscript. It seems that the authors also responded carefully to the comments of the two other reviewers.

I am particularly happy to see that the authors now consider a more realistic solidus curve in their calculations and that their original conclusions still hold.

I have no issues with the manuscript as it is. I think it is a substantial contribution to our understanding of Mercury's volcanic history and the formation of the crust.

My only comment is still about the title. It could still be improved. I'm still not sure that it is easy to understand what the authors mean by 'indicative of thermal state and composition'

Dear Reviewer,

Thank you again for your feedback. Following yours and S. Marchi's comments we changed the title, which now should better reflect the main findings of the paper.

Sincerely,

Sebastiano Padovan

REVIEWERS' COMMENTS:

Reviewer #1 (Remarks to the Author):

The authors have sufficiently addressed my comments and I think the manuscript is now in good shape to be accepted.

One minor comment: The authors have forgotten to replace "Regolith" with "Megaregolith" in the figures (e.g., Figs. 1, 2 etc).